# Structural basis of the interaction between SETD2 methyltransferase and hnRNP L paralogs for governing co-transcriptional splicing

Saikat Bhattacharya [1,5], Suman Wang[2,3,5], Divya Reddy[1], Siyuan Shen[2,3], Ying Zhang[1], Ning Zhang[1], Hua Li[1], Michael P. Washburn [4], Laurence Florens [1], Yunyu Shi[2,3], Jerry L. Workman [1✉] & Fudong Li[2,3✉]

The RNA recognition motif (RRM) binds to nucleic acids as well as proteins. More than one such domain is found in the pre-mRNA processing hnRNP proteins. While the mode of RNA recognition by RRMs is known, the molecular basis of their protein interaction remains obscure. Here we describe the mode of interaction between hnRNP L and LL with the methyltransferase SETD2. We demonstrate that for the interaction to occur, a leucine pair within a highly conserved stretch of SETD2 insert their side chains in hydrophobic pockets formed by hnRNP L RRM2. Notably, the structure also highlights that RRM2 can form a ternary complex with SETD2 and RNA. Remarkably, mutating the leucine pair in SETD2 also results in its reduced interaction with other hnRNPs. Importantly, the similarity that the mode of SETD2-hnRNP L interaction shares with other related protein-protein interactions reveals a conserved design by which splicing regulators interact with one another.

[1] Stowers Institute for Medical Research, Kansas City, MO 64110, USA. [2] Hefei National Laboratory for Physical Sciences at Microscale, School of Life Sciences, Division of Life Sciences and Medicine, University of Science and Technology of China, Hefei, Anhui 230026, China. [3] Ministry of Education Key Laboratory for Membraneless Organelles and Cellular Dynamics, University of Science and Technology of China, Hefei, China. [4] Department of Cancer Biology, University of Kansas Medical Center, Kansas City, KS 66160, USA. [5] These authors contributed equally: Saikat Bhattacharya, Suman Wang. ✉email: jlw@stowers.org; lifudong@ustc.edu.cn

Alternate splicing (AS) is a deviation from the more prevalent process of splicing in which certain exons are skipped resulting in various forms of mature mRNA[1]. AS is a vital process that enables cells to synthesize multiple protein isoforms from the same gene. Totally, 95% of human genes are estimated to undergo AS and it gives rise to the protein diversity needed for the varied cell types and functions from a limited set of genes[2,3]. AS functions in critical biological processes including cell growth, cell death, cell differentiation pluripotency, and development[4,5]. Defects in AS cause neurodegenerative diseases and cancer[6–10].

The exons that end up in the mature mRNA during the process of AS are defined by the interaction between *cis*-acting elements and *trans*-acting factors. *Cis*-acting elements include exonic- and intronic-splicing enhancers (ESEs/ISEs) that are bound by positive trans-acting factors, such as SR (serine/arginine-rich) proteins, and exonic- and intronic-splicing silencers (ESSs/ISSs) are bound by negative trans-acting factors, such as the heterogeneous nuclear ribonucleoproteins (hnRNPs)[1,11,12]. The collaboration between these elements results in the promotion or inhibition of spliceosome assembly on weak splice sites. hnRNPs are highly conserved from nematodes to mammals and have several critical roles in mRNA maturation[13,14]. Their function during AS is to bind to the ESS and can compete with the SR proteins for binding.

Studies over the years have largely focused on the nucleic acid binding aspect of hnRNPs to understand their function and the mechanism of AS. However, analysis of the RNA binding motif of hnRNPs has revealed that their binding regions are widespread in mRNAs. For example, studies aimed to find the RNA binding motif of hnRNP L revealed that it binds to CA-rich regions[15–19]. Such sequences, however, occur in the human genome at a frequency of 19.4 CA repeats per megabase[20], representing the most common simple sequence repeat motif. Combined with the fact that hnRNPs are very abundant and ubiquitous proteins, this makes it unclear how the hnRNPs engage their target transcripts specifically[21]. Also, while a context-dependent regulation of splicing by hnRNP L has been noted, it is unknown what factors determine this process[22].

One possibility may be that hnRNPs rely on their interacting protein partners to engage their specific target pre-mRNA in a context-dependent and cell line-specific manner. The evidence to back this possibility comes from the study of the interactome of hnRNP L. We and others have shown that it specifically interacts with the methyltransferase SETD2 and the mediator complex component Med23[23,24]. Both SETD2 and Med23 are functionally important proteins. SETD2 deposits the conserved H3K36me3 mark besides methylating substrates such as STAT1 and tubulin[25,26]. These activities make SETD2 a regulator of DNA repair, alternative splicing, and DNA methylation[27–31]. Med23 is part of the Tail module of the Mediator complex, a central integrator of transcription[32]. Med23 connects the complex to sequence-specific transcription factors[33]. Med23 brings hnRNP L to the promoter of target genes from where it might be handed over to SETD2 to coregulate a common subset of AS events. Notably, SETD2 binds to RNA Pol II during transcription elongation. Therefore, the hnRNPs interactome not only regulates its function but also couples transcription and alternative splicing, which permits the sequential recognition of emerging splicing signals by the splicing machinery.

The RNA recognition motifs (RRMs), specifically the RRM2 of hnRNP L, mediate its interaction with other proteins. We recently showed that the RRM2 of hnRNP L binds to a novel SETD2–hnRNP Interaction (SHI) domain in SETD2[23]. Consistent with the focus on the RNA binding aspect of hnRNPs, numerous crystal structures of RRM–RNA complexes are available[34–36]. However, the molecular basis for the specific binding of protein interactors by hnRNPs remains elusive.

Furthermore, it is not clear why the RRM2 is specifically able to interact with SETD2 but the other RRMs of hnRNP L cannot.

Notably, hnRNP L has a paralog hnRNP LL that has a very similar amino acid sequence. Although they have similar specificity for RNA substrate recognition, their RNA-binding constraints are different and they have been shown to have non-redundant roles in regulating AS[37,38]. For example, hnRNP L and LL both bind to the same regulatory element in exon 4 of the CD45 gene, but hnRNP LL induces more repression than hnRNP L[39–42]. Also, tissue-specific differences in the expression of these paralogs correlate with their distinct functions. hnRNP LL expression is high in testes. It also increases significantly during B cell to plasma cell differentiation and T cell activation[13,43,44]. It remains to be tested whether SETD2 can directly bind hnRNP LL.

In this work, we report the crystal structure of the hnRNP L-RRM2 in complex with the SETD2-SHI domain at 1.80 Å resolution and show that a leucine pair in the disordered region of SETD2 forms hydrophobic interactions with RRM and is crucial for binding. Notably, the SETD2 binding region in RRM2 is distinct from its RNA-binding interface. Furthermore, we demonstrate that the hnRNP L paralog, hnRNP LL, also interacts with SETD2. SETD2 co-purifies numerous RNA binding proteins besides hnRNP L/LL. Strikingly, mutating the two conserved leucines in the SETD2 SHI domain results in loss of its interaction with most RNA-binding proteins. Moreover, our findings reveal that the mode of SETD2-hnRNP L interaction shares similarity with RAVER1-PTB interaction, pointing toward a possible common design behind protein binding by RRMs.

## Results

**A conserved region within the SETD2 SHI domain mediates hnRNP L binding.** Previously, we performed a detailed characterization of SETD2-hnRNP L interaction and demonstrated that the two proteins co-regulate the AS of a subset of genes[23]. This characterization revealed a novel SHI domain in SETD2 that engages the hnRNPs. However, the underlying mechanism of the interaction between them has not yet been characterized. The SHI domain is located in a predicted disordered region in SETD2 and is expected to assume a random coil structure based on ab initio structure modeling[23]. We performed sequence analysis of the SHI domain to look for functional regions using ConSurf[45]. Multiple sequence alignment followed by HMMER homolog search algorithm[46] revealed that the residues 2167–2192 within the 50 residue SHI domain are highly conserved across different organisms (Fig. 1a). Notably, all these species also code for hnRNP L. We wondered whether this conserved stretch is responsible for hnRNP L binding.

To gain insights into this we assayed the binding affinities of SETD2 fragments to hnRNP L by performing isothermal titration calorimetry (ITC). The results showed that SETD2[2167–2192] binds to hnRNP L RRM2 with a mean equilibrium dissociation constant ($K_D$) of $5.86 \pm 0.05\ \mu M$ and an $N$ value of 0.93 (Fig. 1b, Supplementary Tables 1 and 2). Also, we could not detect any interaction between an adjacent fragment SETD2[2113–2140] and hnRNP L RRM2 as expected (Fig. 1c, Supplementary Tables 1 and 2).

Next, we created deletion mutants of SETD2C (1404–2564) in which 10 amino acid bins were deleted in the 50 amino acid SHI domain (Fig. 1d). Previously we have shown that full-length SETD2 protein is robustly degraded[47,48] and hence, the C-terminal region was used for performing purifications. Subsequently, Halo-tagged WT SETD2C and its mutants were affinity-purified using Halo ligand-conjugated magnetic resin from 293 T extracts. The purified complexes were analyzed by silver staining and western blotting (Fig. 1e, f). Consistent with our previous report, hnRNP L was co-

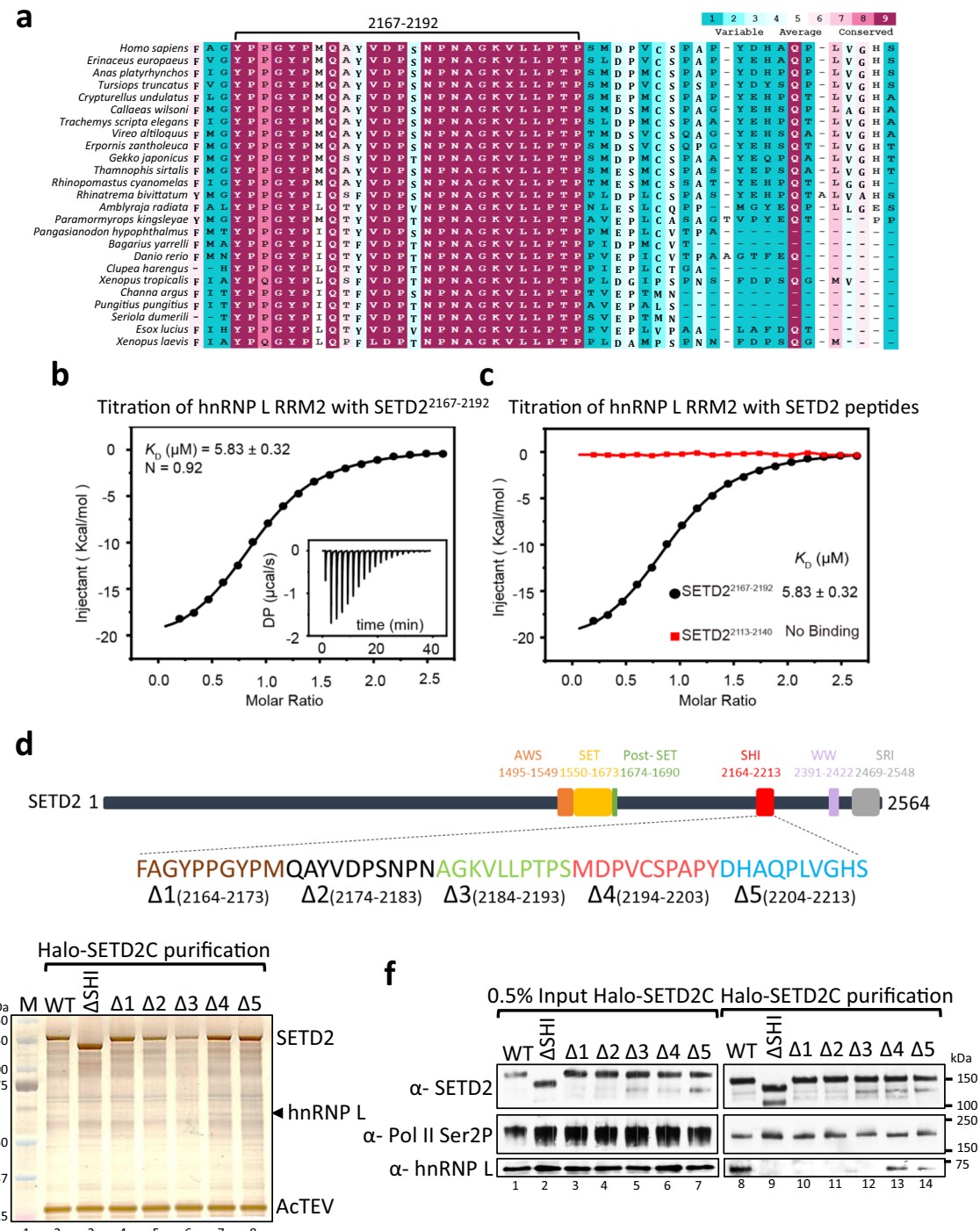

**Fig. 1 A conserved stretch in SETD2 is responsible for its interaction with hnRNP L. a** Multiple sequence alignment of previously identified SHI domain in SETD2. **b** Exemplary ITC titration data of hnRNP L RRM2 with SETD2$^{2167-2192}$ and its fitting curve are shown. $K_D$ dissociation constant, DP differential power, N binding stoichiometry. For the arithmetic mean of $K_D$ values of the three independent experiments and the thermodynamic parameters see Supplementary Table 1. All ITC binding curves are shown in Supplementary Table 2. **c** ITC fitting curves of hnRNP L RRM2 with SETD2$^{2167-2192}$ (black) and SETD2$^{2113-2140}$ (red) are shown. **d** Illustration showing the deletions in SETD2 SHI domain that were made to perform affinity purification. **e** Silver staining and **f** western blotting of affinity-purified complexes of SETD2C and its mutants. The experiment was repeated three times all yielding similar results. Source data are provided as a Source Data file.

purified with SETD2C and the interaction was lost upon deletion of the SHI domain (SETD2CΔSHI) (Fig. 1e, f). Notably, the SETD2CΔSHI 4 and 5 did not affect co-purification of hnRNP L whereas any one of CΔSHI 1–3 (spanning 2164–2193) completely abolished interaction with hnRNP L demonstrating that the stretch 2167–2192 is indeed responsible for hnRNP L binding (Fig. 1e, f).

RNA Pol II which binds to the distinct SRI (Set2–Rpb1 Interaction) domain of SETD2 was co-purified with both WT SETD2C and its SHI domain mutants as expected (Fig. 1f).

To conclude, we identified a conserved 30 amino acid long stretch (2167–2192) in SETD2 that mediates interaction with hnRNP L.

**The crystal structure reveals the details of the hnRNP L-SETD2 interaction.** The conserved 30 amino acid sequence of the SHI domain did not reveal any clues as to how it might engage hnRNP L. Hence, to further explore the molecular basis of this interaction, we determined the crystal structure of hnRNP L RRM2 in complex with the SETD2$^{2167-2192}$ peptide at 1.80 Å resolution. There are two complexes contained in a crystallographic asymmetric unit, with hnRNP L RRM2 chains A and C associating with SETD2$^{2167-2192}$ chains B and D, respectively (Supplementary Fig. 1a, Supplementary Table 3). The two complex pairs are highly similar to each other, with root-mean-square deviation (RMSD) of 0.357 Å of overall 101 aligned Cα atoms. The hnRNP L RRM2 in the complex adopts a β1α1β2β3α2β4β5 conformation that forms a five-stranded β sheet packed against two α-helices. Further, three-dimensional structure superimposition of mouse apo hnRNP L RRM2 (PDB ID: 2MQM, sharing 100% sequence identity with human hnRNP L RRM2) and our structure (RMSD = 0.672 Å over 92 aligned Cα atoms) reveals no significant structural changes in RRM2 upon SETD2$^{2167-2192}$ peptide binding (Supplementary Fig. 1b).

When bound to hnRNP L RRM2, SETD2$^{2167-2192}$ adopts a U-shaped conformation contacting the RRM dorsal helical face (Fig. 2a). Although all residues of the SETD2$^{2167-2192}$ peptide could be clearly traced, only a fraction at the C-terminus of this peptide ($^{2183}$NAGKVLLPTP$^{2192}$) was found to directly contact the RRM, burying ~477 Å of the solvent-exposed area of hnRNP L RRM2. In detail, the bulk side chain of SETD2$^{Leu2188}$ is accommodated by a hydrophobic pocket which is lined by hnRNP L$^{Leu251}$ from α2, hnRNP L$^{Ile214}$ from α1, hnRNP L$^{Ile256}$ from the α2-β4 loop, and hnRNP L$^{Leu263}$ from β4 (Fig. 2b–d). In addition, the side chain of SETD2$^{Leu2189}$ stretches into a neighboring shallow apolar depression lined by hnRNP L Ile214, Val210, Ile256, and Tyr257 (Fig. 2b–d). Furthermore, the residues downstream of the Leu pair, including 2190-Pro-Thr-Pro-2192, together with SETD2$^{Leu2189}$, wrap around the hnRNP L$^{Tyr257}$ and hnRNP L$^{Tyr204}$, providing further binding affinity and specificity (Fig. 2e). Notably, hnRNP L$^{Tyr257}$ makes van der Waals contacts with both SETD2$^{Leu2189}$ and SETD2-$^{Pro2192}$, playing an important role in hnRNP L-SETD2 interaction. Besides apolar contacts, there are also substantial hydrogen bonding interactions between SETD2 peptide and hnRNP L RRM2. The amide nitrogen of SETD2$^{Leu2188}$ and SETD2$^{Leu2189}$ form hydrogen bonds with the carbonyl oxygen of hnRNP L$^{Asp255}$ and hnRNP L$^{Ile256}$, respectively (Fig. 2f). In addition, the carbonyl oxygen and N$_\zeta$ atom of SETD2$^{Lys2186}$ make hydrogen bonding interactions with hnRNP L$^{Asp255}$ and hnRNP L$^{Ser250}$, respectively (Fig. 2f). Furthermore, the side chain of SETD2$^{Asn2183}$ forms bifurcated hydrogen bonds with the main chain of hnRNP L$^{Ala249}$ and hnRNP L$^{Asn252}$ (Fig. 2f). Notably, the structure also revealed that hnRNP L RRM2 can form a ternary complex with SETD2 and RNA by binding them simultaneously as their binding sites are non-overlapping (Supplementary Fig. 1c).

**hnRNP LL and SETD2 interact in vivo.** Previously, we showed that ingenuity pathway analysis (IPA) of proteins co-purified with SETD2 revealed enrichment of RNA processing proteins, including numerous hnRNPs[23]. HnRNP LL has a very similar amino acid sequence with 68% sequence identity and domain organization to its paralog hnRNP L (Fig. 3a). Despite being paralogous, hnRNP L and LL have different RNA-binding constraints and have been shown to have a non-redundant role in regulating AS[37,38]. Notably, unlike hnRNP L, hnRNP LL did not interact with Med23 in vitro and did not interact much in the

coimmunoprecipitation experiment[24]. We wanted to test whether SETD2 and hnRNP LL also interact in vivo.

Multi-dimensional protein identification technology (MudPIT) mass spectrometry analysis of purified complexes with SETD2C indeed revealed hnRNP LL as an interactor (Fig. 3b). Moreover, MudPIT analysis of purified SETD2C complexes from 293 T cell extracts depleted of hnRNP L revealed enrichment of hnRNP LL suggesting that it binds to SETD2 independent of hnRNP L (Fig. 3c). In line with our previous finding that SETD2–hnRNP L interaction occurs irrespective of SETD2–Pol II interaction, mass spectrometry analysis revealed that SETD2–hnRNP LL interaction persisted even upon the deletion of the SRI domain from SETD2C (Fig. 3b, d). To confirm the mass spectrometry results, western blotting with an antibody specific for hnRNP LL was performed with affinity-purified complexes using Halo-SETD2CΔSRI as bait from 293 T extracts with and without RNase treatment. In this technique, elution of the proteins purified involves cleaving off the Halo-tag with TEV protease, thus, resulting in a difference in molecular weight of bait between input and eluted samples. The results confirmed that SETD2 interaction with hnRNP LL persisted even without the Pol II interaction domain and upon RNase treatment (Fig. 3e). Furthermore, mass spectrometry analysis of SETD2CΔSHI mutant revealed loss of interaction with hnRNP LL strongly suggesting that it binds to the same region in SETD2 as hnRNP L (Fig. 3b, d). To validate the mass spectrometry results, SETD2 deletion mutant SETD2CΔSHI$^{2164-2213}$, was affinity-purified from 293 T cells and analyzed by immunoblotting. As anticipated, immunoblotting for RNA Pol II and anti-hnRNP LL revealed that upon deletion of the stretch 2164–2213 from SETD2, the SETD2–hnRNP LL interaction was abolished without affecting the SETD2–Pol II interaction (Fig. 3f). Further, the mass spectrometry analysis of purification of SETD2 2164–2213 revealed hnRNP LL as an interactor demonstrating that hnRNP LL binds to the SETD2 SHI domain (Fig. 3g).

Mass spectrometry analysis of yeast SETD2 homolog, Set2 (ySet2) purified from 293 T cells revealed that it can interact with Pol II even in human cells and this interaction was lost upon the deletion of the SRI domain as expected (Fig. 3h, i). However, an interaction between ySet2 and hnRNP LL was not observed consistent with the fact that although ySet2 and SETD2 share the conserved AWS, SET, Post-SET, WW, and SRI domains, ySet2 lacks the SHI domain (Fig. 3h). Remarkably, the addition of the 2164–2213 stretch of SETD2 to ySet2 (ySet2 2164–2213), and ySet2ΔSRI resulted in a gain of interaction with hnRNP LL (Fig. 3i). These mass spectrometry findings were confirmed by immunoblotting the Set2 purified complexes with an anti-hnRNP LL antibody (Fig. 3j). To conclude, hnRNP LL binds to the SHI domain of SETD2 in vivo.

**hnRNP LL RRM2 domain directly interacts with SETD2.** Based on sequence homology between hnRNP L and LL (71%), RRM2 of LL is expected to interact with SETD2 (Fig. 4a). To test this, Halo-SETD2C and mCherry-HA-hnRNP L/LL RRM2 constructs were co-expressed in 293 T cells and protein complexes were purified using Halo affinity-purification. Immunoblotting of the purified complexes with anti-SETD2 and anti-HA antibodies demonstrated that similar to hnRNP L, the RRM2 of hnRNP LL interacts with SETD2 (Fig. 4b). These findings were confirmed by reciprocal co-purification in which Halo-HA-hnRNP LL RRM2 successfully co-purified GFP-FLAG-SETD2C (Fig. 4c).

To confirm the specificity of SETD2–hnRNP LL RRM2 interaction, hnRNP LL full-length (FL, 1–542) and 272–542 were tagged with mCherry-HA (Fig. 4d). Next, Halo-SETD2C and mCherry-HA-hnRNP LL constructs were co-expressed in 293 T cells and protein complexes were purified using Halo affinity-purification.

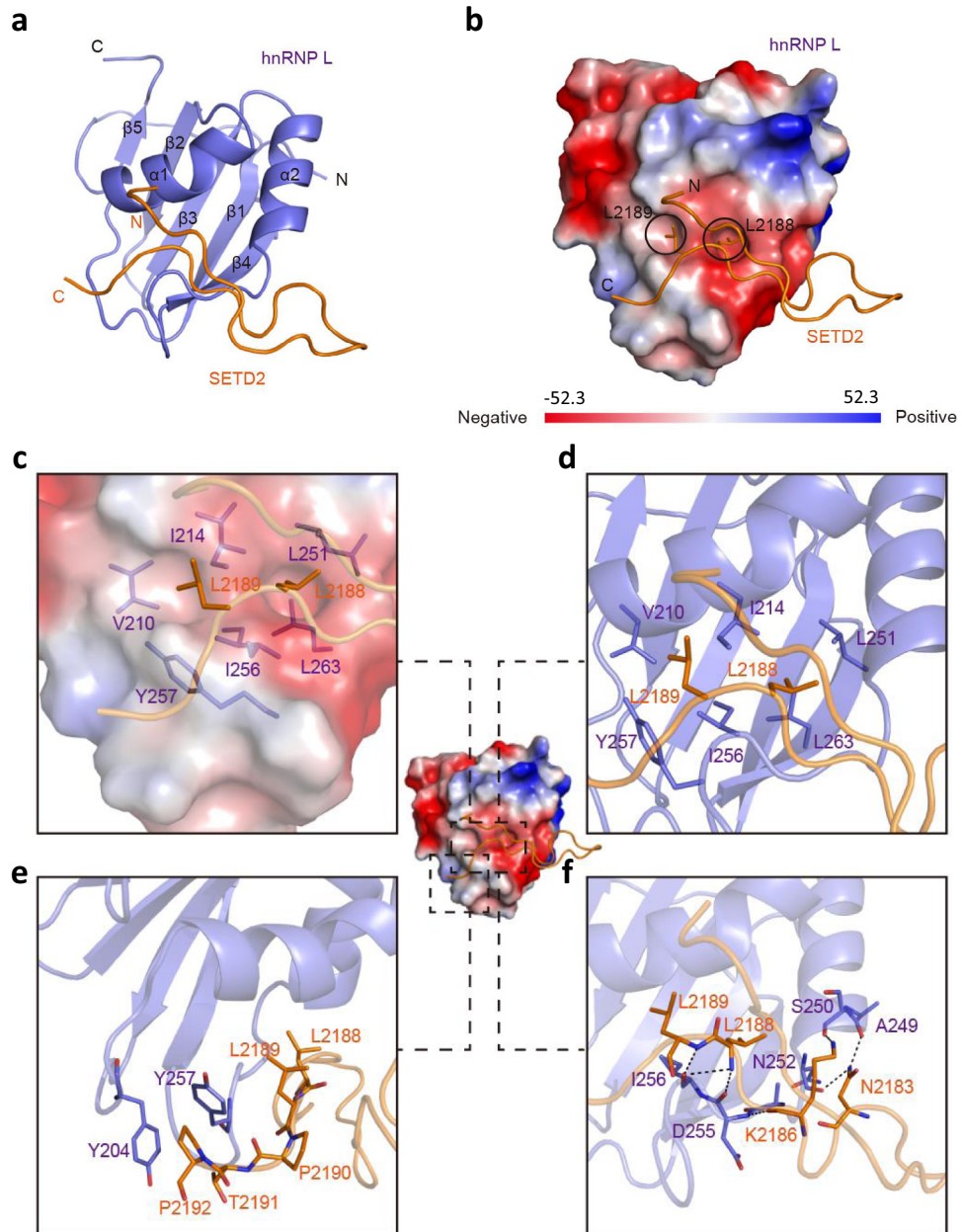

**Fig. 2 The crystal structure reveals the molecular basis of hnRNP L-SETD2 interaction. a** Ribbon representations of hnRNP L RRM2 bound to the SETD2$^{2167-2192}$ peptide. hnRNP L is colored in purple and the bound SETD2$^{2167-2192}$ peptide is colored in orange. **b** The SETD2$^{2167-2192}$ peptide is represented as ribbons on the molecular face of hnRNP L RRM2. The sidechains of SETD2$^{Leu2188}$ and SETD2 $^{Leu2189}$ are shown as sticks. Red and blue colors denote negative and positive surface charges, respectively. The electrostatic potential surfaces were generated with PyMol at the contouring value of the potential from −52.3 to 52.3 kTe$^{-1}$. **c–f** Closeup views of the interactions between hnRNP L and the SETD2$^{2167-2192}$ peptide. (Left) The van der Waals surface views of hnRNP L-SETD2$^{2167-2192}$. hnRNP L (purple) and SETD2$^{2167-2192}$ (orange) are shown as ribbons with selected sidechains as sticks. The van der Waals surface of the hnRNP L is depicted as a semitransparent skin. The SETD2 peptide is represented as a stick diagram (orange). (Right) Hydrogen bonds are shown as black dashed lines.

Immunoblotting with anti-HA antibody demonstrated that hnRNP LL 1-542 interacts with SETD2 as expected, whereas hnRNP LL 272-542 does not (Fig. 4e). Immunoblotting with an anti-Pol II antibody confirmed that Pol II was co-purified (Fig. 4e). These results demonstrate that the RRM2 of hnRNP LL interacts with SETD2 in vivo.

Next, to confirm direct physical interaction, we performed ITC of hnRNP LL RRM2 with SETD2. The ITC result showed that hnRNP LL binds to SETD2$^{2167-2192}$ with a mean $K_D$ value of 8.27 ± 0.28 μM, which is slightly weaker than that of hnRNP L (higher $K_D$ value suggests lower binding affinity) (Fig. 4f,

Supplementary Table 1). In the sequence alignment of RRM2 of hnRNP L vs. LL, we noted that the residues belonging to the SETD2 binding surface are highly conserved, except that I214 of hnRNP L corresponds to a Val in hnRNP LL (Fig. 4a). Next, we generated a hnRNP LL$^{V188I}$ mutant and performed ITC and found that it shows nearly the same binding affinity with SETD2$^{2167-2192}$ as that of hnRNP L (Supplementary Fig. 2a, Supplementary Table 1).

We next tried to crystallize the hnRNP LL RRM2–SETD2$^{2167-2192}$ complex but did not succeed. However, we got a high-resolution structure when a truncated peptide

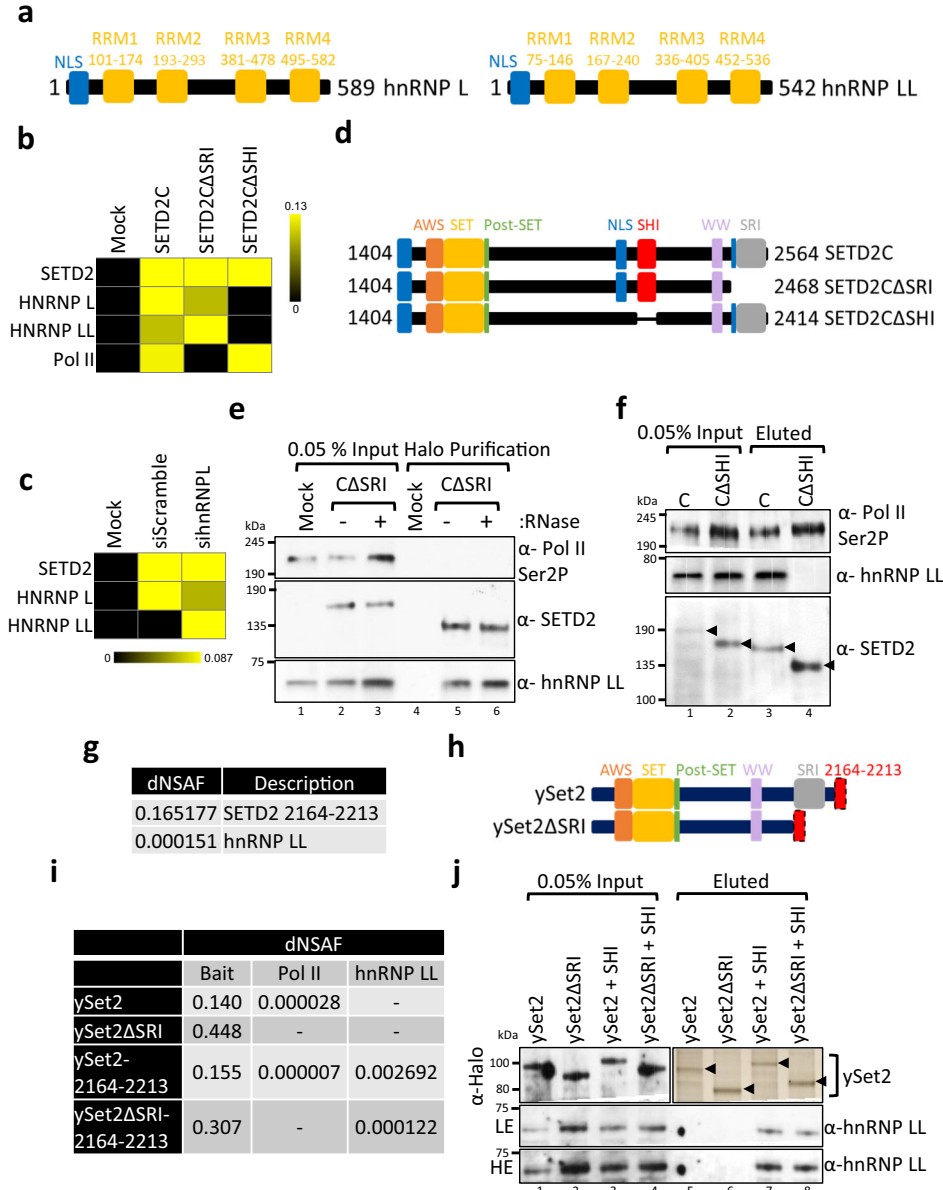

**Fig. 3 SETD2 binds to hnRNP LL through its SHI Domain. a, d, h** Cartoon illustrating the domain organization of hnRNP L, hnRNP LL, SETD2, and ySet2. **b, c** Heat maps showing the enrichment of proteins normalized to the bait (SETD2C) in MudPIT analysis. **e, f, j** Western blotting and silver staining of affinity-purified complexes of Halo-SETD2C or Halo-ySet2 and its mutants from 293 T extracts. The experiment was repeated at least three times all yielding similar results. Source data are provided as a Source Data file. **g, i** Table showing the dNSAFs of the listed proteins post mass spectrometry analysis of purified complexes obtained by affinity purification of Halo-SETD2 or ySet2 from 239 T extracts. AWS associated with SET, SET-Su(var)3-9 enhancer-of-zeste and trithorax, SRI–Set2–Rpb1 interaction, SHI– SETD2–hnRNP interaction dNSAF distributed normalized spectral abundance factor, NLS nuclear localization signal.

encompassing the core interacting motif of SETD2 $^{2180}$SNPNAGKVLLPTP$^{2192}$ was used for crystallization. We determined the structure in the P1 space group at a 1.60 Å resolution, with two almost similar hnRNP LL RRM2-SETD2$^{2180–2192}$ complexes in each crystallographic asymmetric unit. HnRNP LL-SETD2$^{2180–2192}$ structure shows high similarity to the structure of hnRNP L- SETD2$^{2167-2192}$complex (RMSD = 0.672 Å over 92 aligned Cα atoms) (Fig. 4g, Supplementary Table 3).

**Mutating key residues in hnRNP L and SETD2 abolishes their interaction**. The crystal structures revealed the important residues in RRM2 that mediate interaction with SETD2. To confirm

that those residues are indeed important for binding SETD2, ITC was performed using a series of hnRNP L mutants. Mutating the residues I256 and Y257 of hnRNP L had the biggest impact on reducing SETD2 binding (Fig. 5a, Supplementary Table 1). The I256A mutation resulted in a ninefold decrease in SETD2 binding whereas the Y257A mutation completely abolished SETD2 binding.

To test whether these mutations also resulted in the loss of hnRNP L-SETD2 binding in vivo, Halo-SETD2C and mCherry-HA-hnRNP L constructs were co-expressed in 293 T cells and protein complexes were purified using Halo affinity-purification. Immunoblotting with anti-SETD2 and anti-hnRNP L revealed successful purification of endogenous hnRNP L with SETD2 as expected (Fig. 5b). Also, immunoblotting with an anti-HA

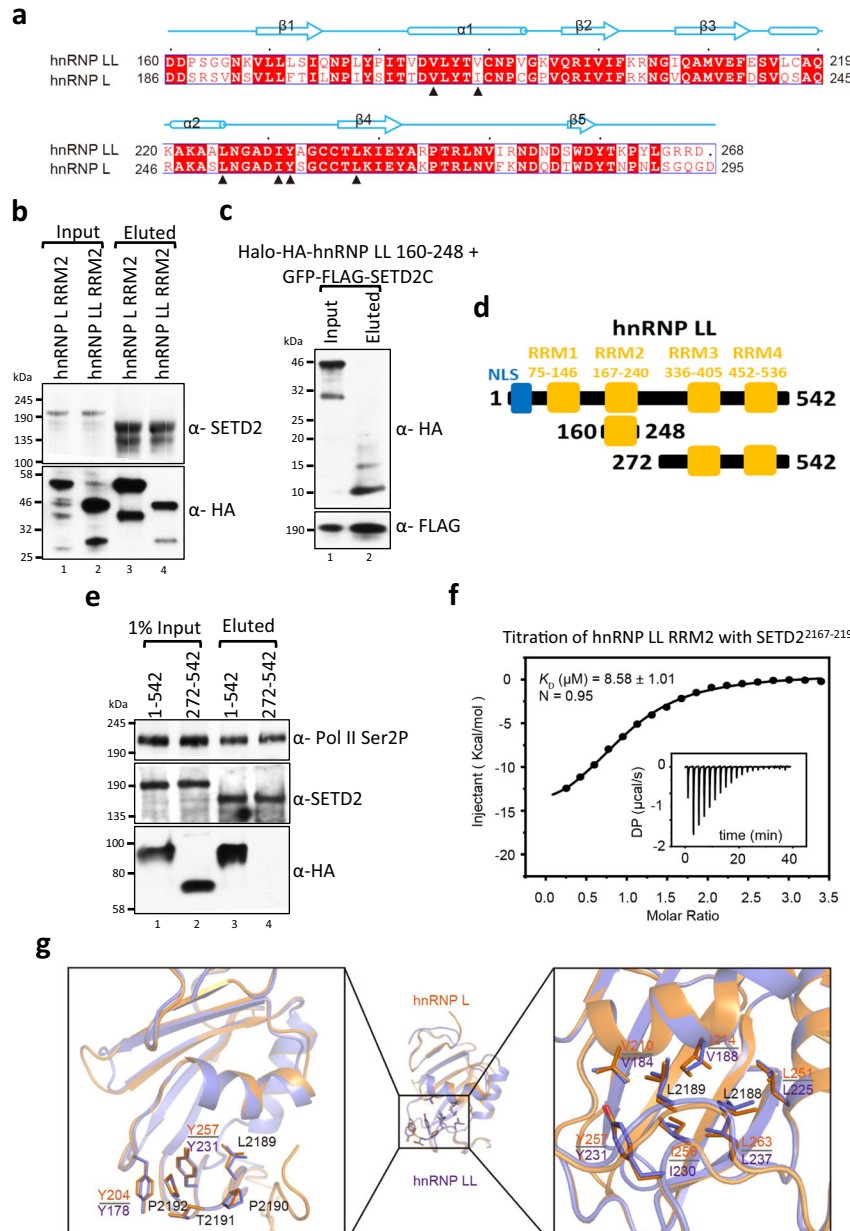

**Fig. 4 hnRNP LL RRM2–SETD2 interaction occurs in a similar fashion as hnRNP L–SETD2 binding. a** Sequence alignment of hnRNP L RRM2 and hnRNP LL RRM2 proteins. The alignment was generated by ESPript3[79] with CLUSTALW[80]. The secondary structures of hnRNP L RRM2, as determined by DSSP, are shown above the sequences. Red squares denote identical residues whereas black triangles highlight the key residues involved in the interaction with SETD2. **b, e** Halo purification was performed from extracts of 293 T cells co-expressing Halo-tagged SETD2C and mCherry-HA-hnRNP L/LL constructs. Input and eluted samples were resolved on gel followed by western blotting. The experiment was repeated at least 2 times all yielding similar results. Source data are provided as a Source Data file. **c** Halo purification was performed from extracts of 293 T cells co-expressing Halo-HA-tagged hnRNP LL and GFP-FLAG-SETD2C constructs. Input and eluted samples were resolved on gel followed by western blotting. The experiment was repeated at least two times all yielding similar results. Source data are provided as a Source Data file. **d** Schematic representation of hnRNP LL segments used in purification experiments. **f** ITC titration data of hnRNP LL with SETD2[2167-2192] and its fitting curve are shown. For the arithmetic mean of $K_D$ values of the three independent experiments and the thermodynamic parameters see Supplementary Table 1. All ITC binding curves are shown in Supplementary Table 2. **g** (Left) Structure comparison of hnRNP L_RRM2–SETD2[2167-2192] complex (orange) and hnRNP LL_RRM2–SETD2[2167-2192] complex (purple). (Right) Two complexes are shown as ribbons with selected sidechains as sticks. $K_D$ dissociation constant, DP differential power, N binding stoichiometry. RRM–RNA recognition motif, NLS nuclear localization signal.

antibody confirmed the co-purification of ectopically expressed WT hnRNP L FL and the lack of SETD2 binding to hnRNP L 322–589 as we have previously shown[23] (Fig. 5b). Importantly, the hnRNP L FL mutant in which both I256 and Y257 were mutated to alanine did not interact with SETD2, confirming our ITC data (Fig. 5b).

Only the RRM2 out of the four RRMs of hnRNP L binds SETD2. To understand this, we analyzed the sequences of hnRNP L RRMs. The residues V210, I214, L251, I256, Y257, L263 of hnRNP L RRM2 form two hydrophobic pockets (Fig. 2c). Sequence alignment shows that the other three RRMs do not contain these six conserved hydrophobic residues, therefore,

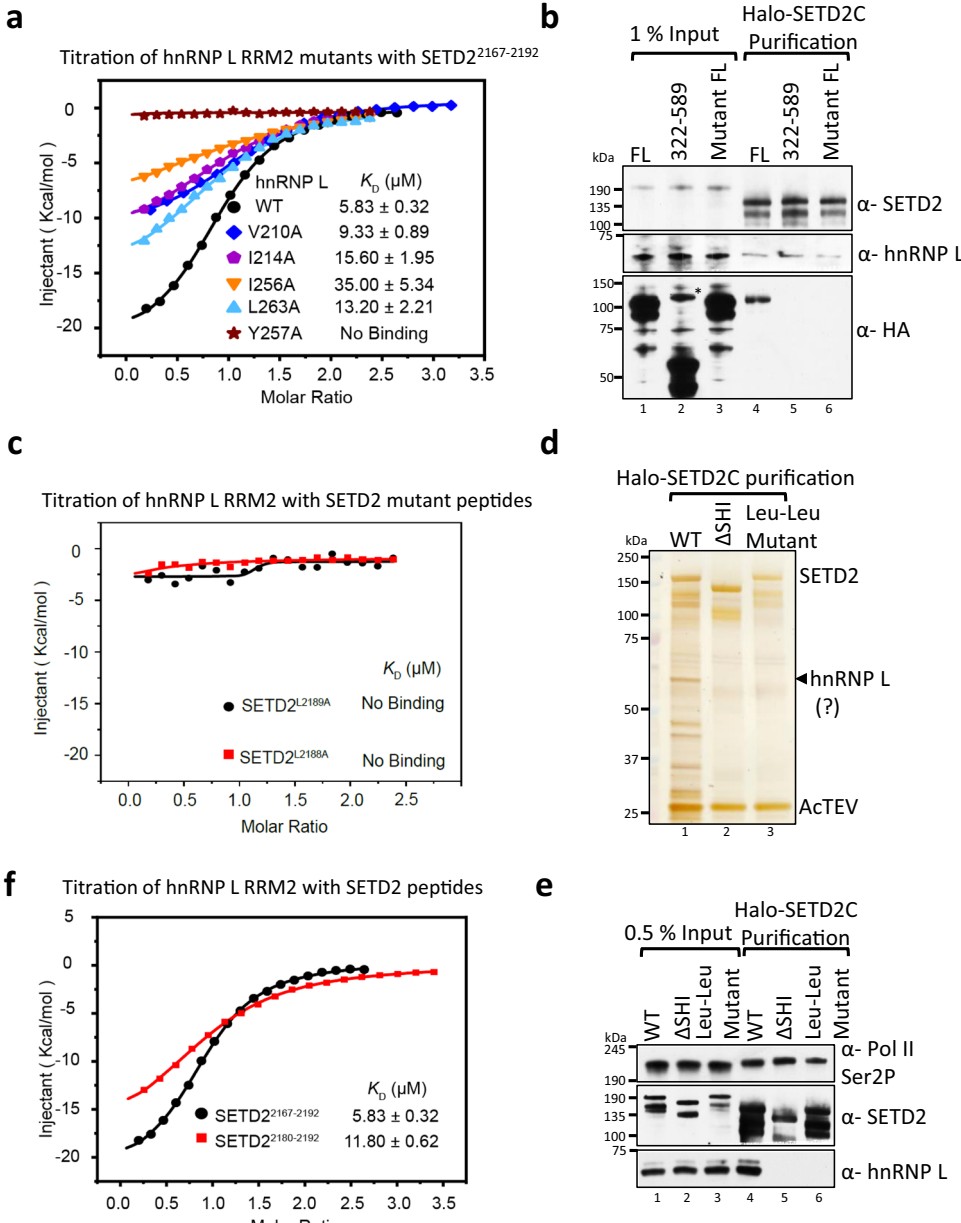

**Fig. 5 Mutating key residues in hnRNP L and SETD2 disrupts their in vitro and in vivo binding. a** ITC fitting curves of hnRNP L RRM2 WT and mutants with SETD2$^{2167-2192}$ peptide are shown. **b**, **d**, **e** Western blotting and silver staining of affinity-purified complexes of Halo-SETD2C WT and its mutants. In (**b**), mCherry-HA-hnRNP L proteins were used as prey. * denotes non-specific band. The experiment was repeated four times all yielding similar results. Source data are provided as a Source Data file. **c**, **f** ITC fitting curves of hnRNP L RRM2 with SETD2 peptides. For the arithmetic mean of $K_D$ values of the three independent experiments and the thermodynamic parameters see Supplementary Table 1. All ITC binding curves are shown in Supplementary Table 2.

SETD2 cannot recognize these three RRMs (Supplementary Fig. 2b).

The crystal structure suggests that the leucine pair in the SETD2 SHI domain is important for binding hnRNP L/LL. Our ITC confirmed this observation as the interaction of SETD2 L2188A and L2189A peptides with hnRNP L RRM2 was completely abolished (Fig. 5c). To test whether these mutations also result in loss of hnRNP L-SETD2 binding in vivo, Halo-SETD2C constructs were expressed in 293 T cells and protein complexes were purified using Halo affinity-purification followed by silver staining and western blotting (Fig. 5d, e). Immunoblotting with anti-SETD2 and anti-hnRNP L revealed that the endogenous hnRNP L co-purified with SETD2C but not with SETDCΔSHI as expected. Importantly, the SETD2 mutant in

which both L2188 and L2189 were mutated to alanine lost interaction with hnRNPL, confirming our ITC data (Fig. 5d, e).

**Residues besides the Leu–Leu in SETD2 are critical for hnRNP L binding**. The mode of interaction between hnRNP L RRM2 and SETD2$^{2167-2192}$ peptide shares resemblance to the mechanism by which RRM2 of PTB, a splicing suppressor, binds to the PRI3 peptide of its co-repressor RAVER1[49]. The two peptides both contain a pair of Leu residues and bind to nearly the same position on the dorsal surface of the RRM (Supplementary Fig. 3a, b). We then asked whether the SETD2$^{2167-2192}$ peptide could also interact with PTB RRM2. The ITC results showed that there is no detectable affinity between SETD2$^{2167-2192}$ and PTB

RRM2, suggesting that the amino acid sequence flanking the Leu–Leu is important for the recognition to occur (Supplementary Fig. 3c, Supplementary Tables 1 and 2). This is consistent with our data in which, besides SETD2CΔSHI 3 that has the Leu–Leu crucial for interaction with hnRNP L, SETD2CΔSHI 1 and 2 also lost interaction with hnRNP L (Fig. 1e, f).

To further ascertain these observations, we performed ITC of hnRNP L RRM2 with a truncated peptide, $^{2180}$SNPNAGKVL LPTP$^{2192}$, belonging to the SETD2 SHI domain. ITC revealed a mean $K_D$ value of $11.57 \pm 0.21\ \mu M$, indicating a weaker binding as compared to SETD2$^{2167-2192}$ and confirming that indeed the N-terminus region of SETD2$^{2167-2192}$ (containing the Δ1 and Δ2 deletions) also plays important role in the binding (Fig. 5f, Supplementary Table 1). Similar results were observed when the binding of hnRNP LL RRM2 with SETD2$^{2180-2192}$ was tested and found to be weaker as compared to binding with SETD2$^{2167-2192}$ (Supplementary Table 1).

**The functions of hnRNP L and LL are partially redundant.**
Previously, we have shown that SETD2 and hnRNP L co-regulate the AS of an overlapping set of events[23]. To test whether SETD2 and hnRNP LL depletion result in overlapping transcriptome changes, RNA-seq has performed post-depleting SETD2 and hnRNP LL in 293 T cells. Depletion of SETD2 and hnRNP LL resulted in significant differential gene expression (FDR < 0.05, fold change > 1.5); 203 differentially expressed genes (57 upregulated, 146 downregulated) were observed upon SETD2 depletion while the expression levels of 514 genes (219 upregulated and 295 downregulated) were altered upon hnRNP LL depletion (Supplementary Fig. 4a, Supplementary Data 1). Furthermore, the analysis of differential AS events revealed that SETD2 and hnRNP LL depletion results in significant changes in 1225 and 1379 AS events, respectively (Supplementary Fig. 4b, Supplementary Data 2). However, the overlap between SETD2 and hnRNP LL regulated gene expression and AS events was very small. For instance, out of the total of 2604 AS changes (FDR < 0.05) brought about by SETD2 and hnRNP LL depletion, only 70 were co-regulated by both (Supplementary Fig. 4b, Supplementary Data 2).

One possible reason behind this observation might be the redundancy between the paralogs hnRNP L and LL. To test the possibility of redundancy in the regulation of transcripts by hnRNP L and LL, RNA-seq was performed post specific depletion of hnRNP L and LL in 293 T cells. The depletion of the targets at the protein level was confirmed by western blotting with antibodies specific for hnRNP L and LL (Supplementary Fig. 5a). Also, hnRNP L depletion did not alter the transcript level of hnRNP LL and vice versa (Fig. 6a). The RNA-seq data revealed a global perturbation in terms of transcription and AS changes upon hnRNP L and LL depletion (Supplementary Data 1 and 2). Strikingly, the overlap of gene expression and AS changes brought about by hnRNP L and LL depletion were very few (Fig. 6b, c, Supplementary Fig. 5b). Moreover, out of the 114 overlapping AS events, almost half of them (43.85%) showed an opposite trend of splicing (Fig. 6d).

Although these results argue for a non-redundant function between hnRNP L and LL, the tenfold higher expression of hnRNP L as compared to hnRNP LL has to be taken into consideration while interpreting these results[13]. This was also reflected in the distributed normalized spectral abundance factor (dNSAF) of these proteins in the MudPIT analysis of SETD2C purification, where hnRNP LL (dNSAF 0.002072) was 28-fold less enriched than hnRNP L (dNSAF 0.058612) (Supplementary Data 3). Therefore, we decided to perform rescue experiments to test whether the differential AS events observed can be specifically rescued by one paralog but not the other. In order to perform

rescue experiments, we first looked for suitable candidate genes. The genes *tjp1* and *bptf* have been reported previously to exhibit differential splicing upon hnRNP L depletion but not hnRNP LL[13]. Our RNA-Seq data confirmed these results as clear retention of exon 20 in *tjp1*- and exon 18a in *bptf* could be seen in genome browser tracks upon hnRNP L depletion but not in control and hnRNP LL depleted cells (Fig. 6e, f). To validate these results, individual AS events were measured by quantitative PCR (qPCR) and represented by the ratios of different exons. Again, the depletion of one paralog did not significantly alter the expression of the other, consistent with our RNA-Seq results (Fig. 6g, Supplementary Fig. 6a). Indeed, hnRNP L depletion led to the increase in retention of exon 20 in *tjp1*, and exon 18a in *bptf* (Fig. 6h–j). Also, no significant change in the ratio of exons was observed upon hnRNP LL depletion consistent with our RNA-Seq data (Fig. 6h–j). Hence, *tjp1* and *bptf* were chosen as candidate genes to test rescue.

For rescue experiments, empty vector, hnRNP L, or hnRNP LL was introduced in 293 T cells depleted of hnRNP L. qPCR revealed specific upregulation of *hnrnpl* and *hnrnpll* as expected (Fig. 6k, Supplementary Fig. 6b). Analysis of AS of *tjp1* and *bptf* revealed that the introduction of mCherry-HA-hnRNP L in hnRNP L depleted cells indeed resulted in the decrease in the ratio of exons, indicating rescue. The expression of mCherry alone did not have that effect despite having a much higher expression (Fig. 6l–n, Supplementary Fig. 6c, d). Strikingly, rescue with mCherry-HA-hnRNP LL also resulted in the decrease in the ratio of exons, suggesting that it could rescue the AS changes brought about by hnRNP L depletion (Fig. 6l–n).

These results show that the functions of hnRNP L and hnRNP LL are at least partially redundant and the non-overlapping transcriptome changes observed upon their depletion might be due to differences in their expression level.

**Mutating Leu–Leu in SETD2 decreases its binding with splicing proteins.** Previously, we have shown that the SHI domain mediates the interaction of SETD2 with numerous RNA processing proteins[23]. We wanted to test whether mutating the Leu–Leu within the SHI domain affects SETD2's ability to bind with other proteins besides hnRNP L and LL. For this, affinity-purified complexes of SETD2C mutants were subjected to MudPIT. A GO-term analysis of the proteins co-purified with the SETD2C revealed enrichment in RNA processing pathways (Fig. 7a). Notably, such enrichment was not observed on the GO-term analysis of co-purified proteins with the SETD2C–Leu–Leu mutant (Fig. 7b).

Next, IPA of the proteins identified through MudPIT revealed that similar to deleting the SHI domain, mutating the Leu https://digiedit3.mpslimited.com/Digicore/DigiEditPage.aspx?FileName=4214022151114994113043556.xmlLeu in the SHI domain led to a significant reduction in the enrichment of protein groups belonging to RNA processing (Fig. 7c). This was not observed upon the deletion of the SRI domain. A closer inspection of the specific proteins associated with such pathways revealed that mutating the Leu-Leu in the SHI domain not only led to the loss of hnRNP L and LL interactions but also resulted in the loss of interaction with other RNA binding proteins (Fig. 7d).

Collectively, the analysis suggests that the two consecutive leucines in the SHI domain are important for the interaction between SETD2 and proteins related to RNA processing.

**Discussion**
It is now clear that AS is coupled to transcription which permits the sequential recognition of emerging splicing signals by the splicing machinery[50]. The finding that hnRNPs such as hnRNP L

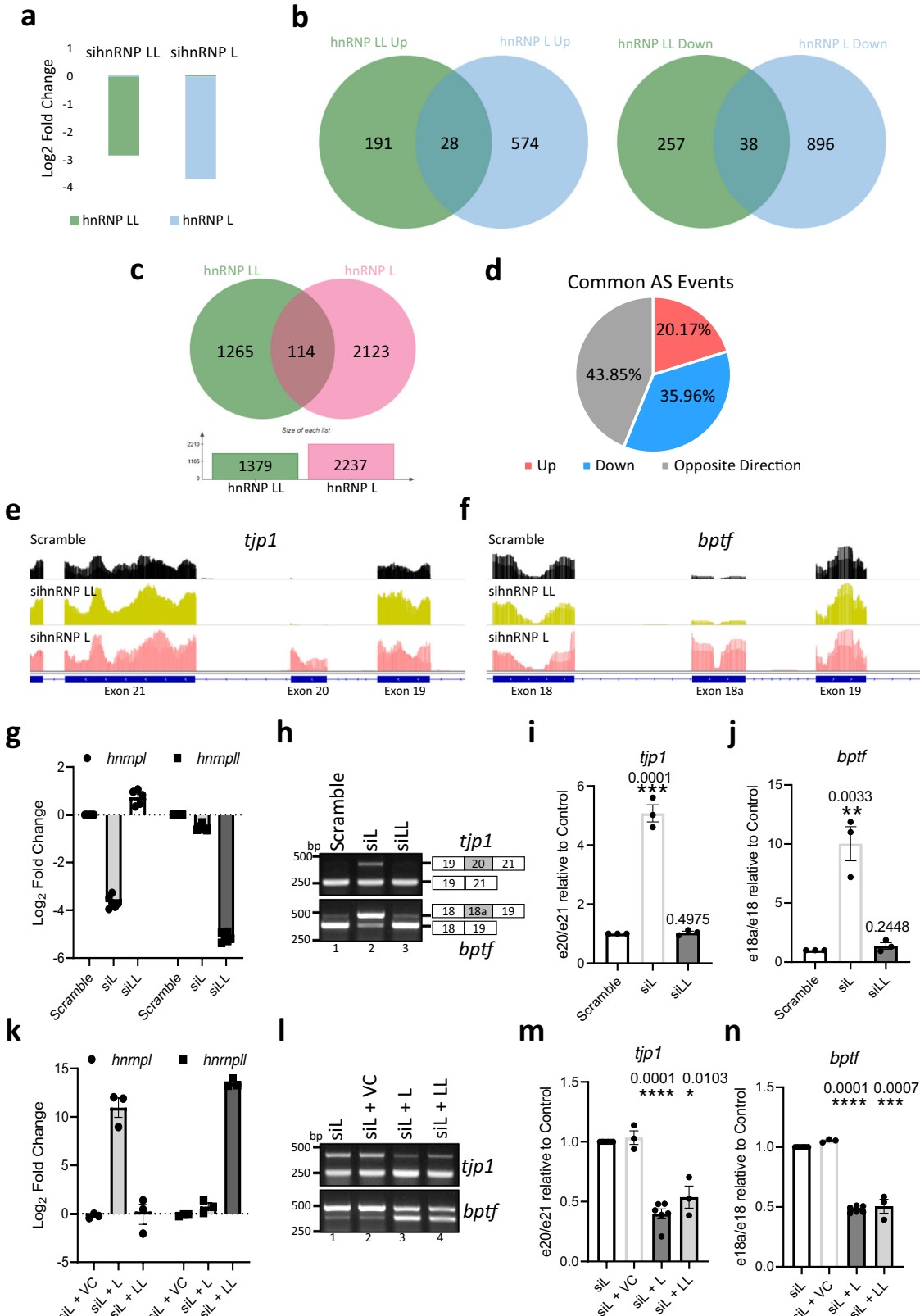

and LL can directly bind to transcription regulators such as Med23 and SETD2 not only shows that they play an important role in such coupling but also signifies the importance of the protein-binding ability of RRMs. Importantly, the non-overlapping binding interfaces for engaging RNA and proteins in hnRNP L RRM2 as revealed by our crystal structure shows that

hnRNP L/LL can form a ternary complex and strongly supports the possibility that the hnRNPs interacting partners aid in their specific recruitment to the target pre-mRNA that we have proposed before[23].

The canonical RRMs are characterized by a β1−α1−β2−β3−α2−β4 structure and the presence of two highly

**Fig. 6 Transcriptome-wide and gene-specific analysis of the regulatory effect of hnRNP L paralogs. a** Chart showing the decrease in expression of the genes depicted based on RNA-seq analysis post siRNA treatment. **b, c** Venn diagrams showing the overlap of differentially expressed genes and AS events upon *hnrnpl* and *hnrnpll* depletion as compared to scramble siRA treated cells. **d** Pie chart showing the fractions of differentially AS events that occur in both *hnrnpl* and *hnrnpll* depletion. **e, f** Genome browser view showing retention of introns in *tjp1* and *bptf* genes upon *hnrnpl* depletion. **g–j** RNA was isolated from 293 T cells 72 h post-transfection with scramble siRNA, or siRNA against *hnrnpl* or *hnrnpll*. Also, **k–n** RNA was isolated from sihnRNPL expressing 293 T cells rescued with vector control (VC), hnRNP L, or hnRNP LL constructs. Specific primers were designed to detect the indicated genes and exons and individual alternative splicing events were checked by RT-PCR (**h, l**) (source data are provided as a Source Data file, the experiment was repeated two times all yielding similar results), measured by quantitative PCR, and represented by the ratios of depicted exons (**i, j, m, n**). For each sample $n = 3$ independent biological samples were examined in the same sequencing run. Data are presented as mean values with a standard error of the mean. An unpaired $t$ test (two-tailed) was performed. $p$-Value $< 0.05$ was considered significant. $p$-Values are depicted on the top of the respective graphs. GAPDH was used for normalization.

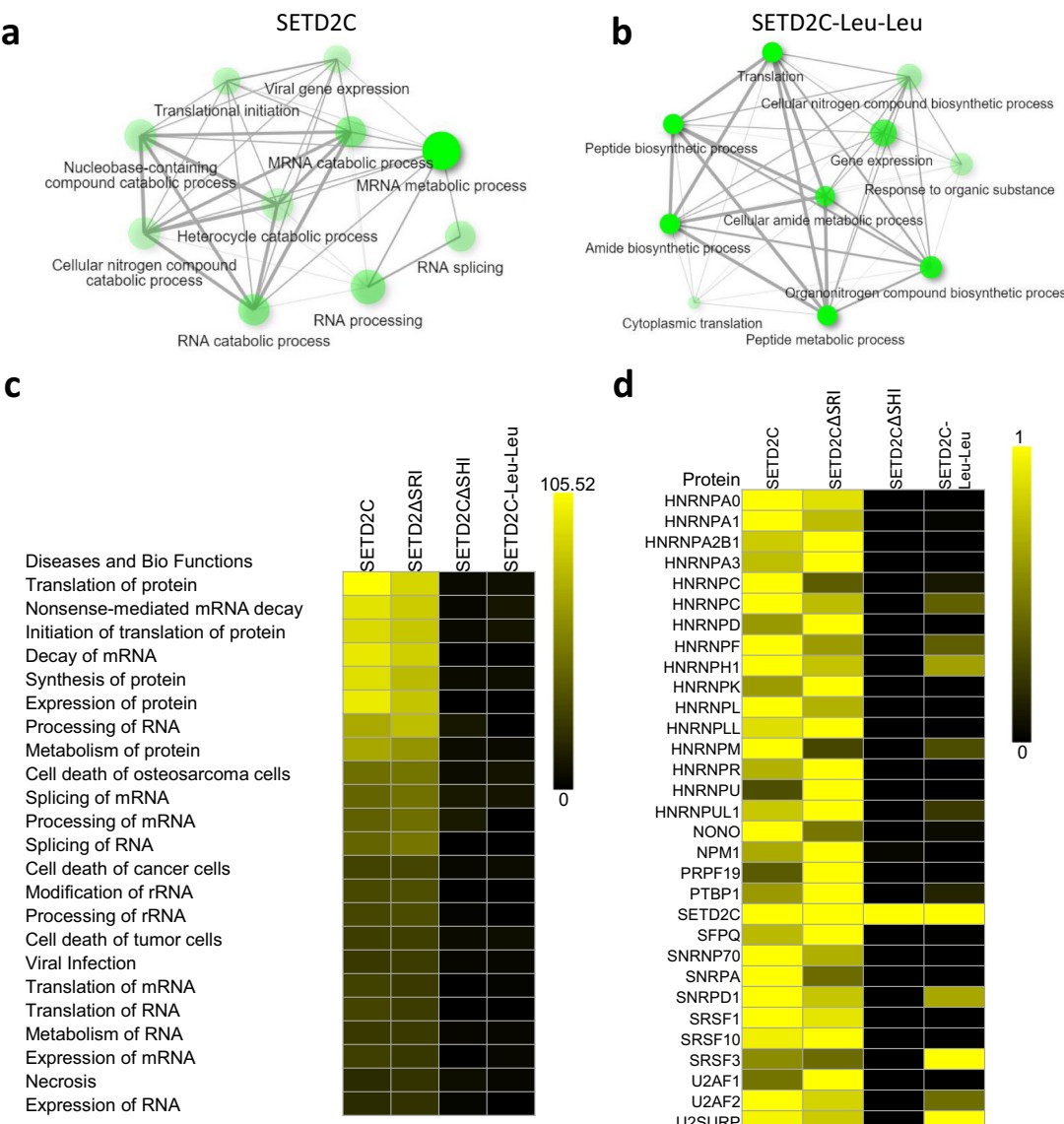

**Fig. 7 Mutating Leu–Leu in the SHI domain results in decreased interaction of SETD2 with RNA-binding proteins. a, b** GO-term analysis of proteins using ShinyGO (http://bioinformatics.sdstate.edu/go/) identified by MudPIT in the affinity-purification of SETD2C and SETD2C-LL mutant. **c, d** Heat maps showing the enrichment of pathways in the IPA (Ingenuity Pathway Analysis) and proteins in MudPIT analysis. The enrichment of proteins is normalized to the bait (SETD2C).

degenerate RNP consensus sequences, RNP-1 and RNP-2[51,52]. The RRM contacts RNA using the RNP-1 and RNP-2 consensus sequences, which are present on the β3 and β1 strands, respectively. This involves primarily hydrophobic interactions between

four conserved aromatic protein side chains and two bases, resulting in the binding of RNA to the β-sheet surface. Our crystal structure revealed that the binding of SETD2 occurs in a very different fashion where its Leu–Leu inserts its side chains into the

two hydrophobic cavities formed by the RRM2 in the opposite face of its RNA binding surface. This mode of interaction is very similar to the way RAVER1 binds PTB. This also shares a resemblance with the way the peptide motif from SF1 binds to U2AF65 RRM3[53]. In this case, the binding of SF1 occurs through the insertion of a tryptophan side chain into a hydrophobic groove on the dorsal face of the RRM. The similarities in the mode of SETD2-hnRNP L, RAVER1-PTB, and SF1-U2AF65 interaction suggest the possibility of a conserved design by which splicing regulators interact with one another.

Besides the Leu–Leu, the N-terminus residues of SETD2 are also important for the binding to occur. Although the N-terminus of the SETD2$^{2167–2192}$ does not interact with the RRM2 directly, we found that the residues from substantial van der Waals and hydrogen contact with the C-terminus of the SETD2 peptide, possibly helping to fix the conformation of the peptide and thus, enhance the binding. Interestingly, a similar phenomenon was found in PTB–RAVER1 interactions, in which a long peptide bound more strongly as compared to the core interacting motif[49]. Also, the flanking region of the core interacting motif belongs to a predicted disordered region in both SETD2 and RAVER1 as per IUPRED2[54] (Supplementary Fig. 3d). This might provide the required flexibility needed to position the core interacting motif for engaging RRM.

Our crystal structure revealed the residues that are important in mediating the interaction between SETD2 and hnRNP L. Notably, some of them are mutated in cancer including the critical residue Y257 of hnRNP L, mutating which led to the abolishment of SETD2–hnRNP L interaction, and P2192 which is part of the LLPTP motif which hnRNP L seems to prefer (Supplementary Fig. 7). We already tested that one such mutation in cancer I214V, a substitution also found in hnRNP LL, results in weaker binding of SETD2. Changes in the expression of both SETD2 and hnRNP L are known to be associated with cancer progression[55–57]. It remains to be seen whether SETD2–hnRNP L interaction or the lack of thereof also affects transformation.

The paralogs hnRNP L and LL have a high sequence identity and RNA-binding specificity. Here we demonstrated that they both engage SETD2. Both are believed to exert their downstream effects in AS by recruiting hnRNPA1[37]. Hence, it is not surprising that they are functionally at least partly redundant. The high expression level of hnRNP L could explain the existence of events that were exclusive to hnRNP L such as *tjp1* and *bptf*, which we confirmed could be rescued by ectopically expressing hnRNP LL also. Moreover, the GO-term analysis of genes showing decreased splicing upon hnRNP L depletion showed enrichment in RNA splicing and mRNA processing pathways (Supplementary Fig. 8a). This was not the case with hnRNP LL (Supplementary Fig. 8b). Moreover, the GO-term analysis of genes showing decreased splicing uniquely upon hnRNP L depletion again showed enrichment in mRNA processing pathways. This makes the comparison between hnRNP L and LL regulated AS events more complicated.

It also must be considered that non-redundancy between these paralogs might exist, especially considering that differences in RNA binding constraints of these proteins have been noted[37]. Moreover, hnRNP LL shows a tissue-specific expression, for instance, high levels in testes, which might suggest a possible tissue-specific role[13]. Regulation of hnRNPs by modulating the expression of their tissue-specific paralogs is known. For instance, the replacement of PTB (hnRNP I) by its paralog, nPTB, which is less repressive for SRC N1 exon splicing than PTB, promotes assembly of an enhancer complex downstream of the exon[58]. Besides, the expression of hnRNP LL increases significantly during B cell to plasma cells differentiation and T cell activation pointing to a context-dependent function[43,44]. It is possible that

in such scenarios where hnRNP LL is more abundant, it might engage SETD2 more preferably than hnRNP L does. It will be interesting to examine in future studies what other proteins the hnRNPs interact with to govern co-transcriptional splicing and the context behind it.

## Methods

**Plasmids.** hnRNP L, hnRNP LL, and SETD2 human ORF were procured from Promega. Deletion mutants of hnRNP L, LL, and SETD2 were constructed by PCR (Phusion polymerase, NEB) using a full-length version of the constructs, respectively as a template, and individual fragments were cloned. All constructs generated were confirmed by sequencing. pCDNA3-ySet2 were procured from Addgene. siRNA for *setd2*, *hnrnpll*, and *hnrnpl* as well as scramble siRNA sequences were procured from Dharmacon.

**Cell line maintenance and drug treatment.** 293 T cells were procured from ATCC and maintained in DMEM supplemented with 10% FBS and 2 mM L-glutamine at 37 °C with 5% CO$_2$. Transfections of plasmids were performed using Fugene HD (Promega) and that of siRNAs was performed using Lipofectamine RNAi Max (Thermosfisher) at 40% cell confluency.

**Affinity purification.** 293 T cells expressing the protein of interest were harvested in 1× PBS and collected by centrifugation. The cells were lysed by resuspending in lysis buffer (50 mM Tris, pH 7.5, 150 mM NaCl, 1% Triton-X 100, 0.1% Na-deoxycholate, and a protease inhibitor cocktail). The lysed cells were centrifuged at 16,000*g* for 20 min. The supernatant was collected and diluted 1:3 by adding dilution buffer (1× PBS, pH 7.5 with 1 mM DTT and 0.005% NP-40). The diluted lysate was added to pre-equilibrated Magne® HaloTag® Beads (Promega, G7282) and incubated overnight on a rotator at 4 °C. The beads were then washed with wash buffer (50 mM Tris-HCL, pH 7.5, 300 mM NaCl, 0.005% NP40, and 1 mM DTT. AcTEV (ThermoFisher, 12575015) protease was used for elution.

**Mass spectrometry analysis.** TCA precipitated protein samples were analyzed independently by Multidimensional Protein Identification Technology (MudPIT)[59,60]. Briefly, precipitated protein samples were resuspended in 100 mM Tris pH 8.5, 8 M urea to denature the proteins. Proteins were reduced and alkylated prior to digestion with recombinant LysC (Promega) and trypsin (Promega). Reactions were quenched by the addition of formic acid to a final concentration of 5%. Peptide samples were pressure-loaded onto 100 μm fused silica microcapillary columns packed first with 9 cm of reverse phase material (Aqua; Phenomenex), followed by 3 cm of 5-μm Strong Cation Exchange material (Luna; Phenomenex), followed by 1 cm of 5-μm C18 RP. The loaded microcapillary columns were placed in-line with a 1260 Quartenary HPLC (Agilent). The application of a 2.5 kV distal voltage electrosprayed the eluting peptides directly into LTQ linear ion trap mass spectrometers (Thermo Scientific) equipped with a custom-made nano-LC electrospray ionization source. Full MS spectra were recorded on the eluting peptides over a 400–1600 m/z range, followed by fragmentation in the ion trap (at 35% collision energy) on the first to fifth most intense ions selected from the full MS spectrum. Dynamic exclusion was enabled for 120 sec[61]. Mass spectrometer scan functions and HPLC solvent gradients were controlled by the XCalibur data system (Thermo Scientific).

RAW files were extracted into.ms2 file format[62] using RawDistiller v. 1.0, in-house developed software[63]. RawDistiller D(g, 6) settings were used to abstract MS1 scan profiles by Gaussian fitting and to implement dynamic offline lock mass using six background polydimethylcyclosiloxane ions as internal calibrants[63]. MS/MS spectra were first searched using ProLuCID[64] with a 500 ppm mass tolerance for peptide and fragment ions. Trypsin specificity was imposed on both ends of candidate peptides during the search against a protein database combining 44,080 human proteins (NCBI 2019-11-03 release), as well as 426 common contaminants such as human keratins, IgGs, and proteolytic enzymes. To estimate false discovery rates (FDR), each protein sequence was randomized (keeping the same amino acid composition and length) and the resulting "shuffled" sequences were added to the database, for a total search space of 89,038 amino acid sequences. A mass of 57.0125 Da was added as a static modification to cysteine residues and 15.9949 Da was differentially added to methionine residues.

DTASelect v.1.9[65] was used in combination with our in-house script, swallow v. 0.0.1 (https://github.com/tzwwen/kite) to control FDRs to less than 1%. Results from each sample were merged and compared using CONTRAST[65]. Combining all replicates, proteins had to be detected by at least two peptides and/or two spectral counts. Proteins that were subsets of others were removed using the parsimony option in DTASelect on the proteins detected after merging all runs. Proteins that were identified by the same set of peptides (including at least one peptide unique to such protein group to distinguish between isoforms) were grouped together, and one accession number was arbitrarily considered as representative of each protein group.

NSAF7[66] was used to create the final reports on all detected peptides and non-redundant proteins identified across the different runs. Spectral and peptide level FDRs were, on average, 0.52 ± 0.41% and 0.39 ± 0.1%, respectively. QPROT[67] was

used to calculate a log fold change and *Z*-score for the samples compared to the mock control.

For instances where there was more than one replicate analyzed by MudPIT, proteins with log fold change > 1 and *Z*-score > 2 were further analyzed in IPA, Qiagen to determine pathways enriched by the bait proteins. For proteins with only one replicate, a ratio was calculated of dNSAF values between sample and mock. For those to be further analyzed in IPA, the dNSAF ratio had to be >2 compared to mock. Pathways were considered significantly enriched with *p*-value < 0.05 (−log10(*p*-value) > 1.3).

**Recombinant protein expression and purification**. The hnRNP L_RRM2 fragment (residues189–286) was amplified from the human brain cDNA library and cloned into a modified pET28a (Novagen) vector without a thrombin protease cleavage site (termed p28a). All the mutants were generated using a MutantBEST kit (Takara) and verified via DNA sequencing. All the proteins were expressed in *Escherichia coli* BL21 (DE3) cells. The cells were cultured at 37 °C in LB medium, until the OD600 reached about 0.8. Then the proteins were induced with 0.2 mM isopropyl β-D-1-thiogalactopyranoside (IPTG). After induction at 16 °C for 24 h, the cells were harvested and lysed by sonication in buffer A (20 mM Tris-HCl, pH 7.8, 1 M NaCl). Proteins were first purified by Ni-NTA agarose beads and further purified by size-exclusion chromatography on a Hiload 16/60 Superdex 75 column (GE Healthcare) in buffer A. Purified proteins were dialyzed with Buffer B (20 mM Tris-HCl pH 7.8, 150 mM NaCl) and concentrated for subsequent analysis.

**Protein complex preparation**. The synthetic SETD2 (residues 2167–2192 and 2180–2192) peptides were dissolved into buffer B, and the peptide was mixed with the purified protein at a 1.5:1 molar ratio and incubated at 16 °C overnight to form a complex. The complex was condensed to 1.0 mM in preparation for crystallization.

**Crystallography**. The crystals were grown at 20 °C via the sitting-drop vapor diffusion method. The crystals of hnRNP L in complex with SETD2[2167–2192] were grown by mixing 1 μL of the protein complex and 1 μL of reservoir buffer (0.8 M potassium/sodium phosphate, pH 7.5). The crystals of hnRNP LL in complex with SETD2[2180–2192] were grown by mixing 1 μL of the protein complex and 1 μL of reservoir buffer (0.3 M Ammonium Sulfate, 20% PEG 4000). All the crystals were harvested in their corresponding reservoir buffers supplemented with 25% (v/v) glycerol and frozen in liquid nitrogen.

X-ray diffraction data sets of the crystals were collected at beamline 19U1 at the Shanghai Synchrotron Radiation Facility with a diffraction wavelength of 0.979 Å. The two data sets, including hnRNP L-SETD2[2167–2192] and hnRNP LL-SETD2[2180–2192], were indexed, integrated, and scaled by the HKL-2000 program suite[68]. The structure of the hnRNP L-SETD2[2167–2192] complex was determined by molecular replacement with the MOLREP 13.07.2020 program using the structure of PTB1-PRI3 (PDB ID: 3ZZY) as the search model. The structure of the hnRNP LL-SETD2[2180–2192] complex was determined by molecular replacement with the MOLREP 13.07.2020 program[69,70] using the structure of hnRNP L-SETD2[2167–2192] as the search model. The model was further built and refined using Coot 0.9.6[71] and Phenix.refine 1.19_4092[72–74], respectively. Crystal diffraction data and refinement statistics are shown in the Supplementary Data. All the structures in the figures were generated using PyMOL 0.99rc6 (DeLano Scientific LLC).

**Isothermal titration calorimetry**. Isothermal titration calorimetry (ITC) assays were performed at 20 °C by using a Microcal PEAQ-ITC instrument (Malvern). We conducted the ITC experiments using SETD2 peptides for titration into hnRNP L and hnRNP LL proteins. SETD2 peptides are added to the syringe at a concentration of about 600 μM and proteins are added to the sample pool at a concentration of about 50 μM. A typical ITC experiment is consisted of 19 drops, with one injection of 1 μL followed by 18 injections of 2 μL of protein sample. The integrated heat data were analyzed using a one-site binding model by MicroCal PEAQ 1.0.0.1259-ITC Analysis Software provided by the manufacturer.

**Isolation of total RNA and PCR**. Total RNA was extracted from cells as per the manufacturer's (Qiagen) instructions. It was further treated with DNaseI (NEB) for 30 min at 72 °C to degrade any possible DNA contamination. RNA (2 μg) was subjected to reverse transcription using QScript cDNA synthesis mix according to the manufacturer's instructions. cDNAs were then amplified with the corresponding gene-specific primer sets. For RTPCR, PCR was conducted for 24 cycles using the condition of 30 s at 94 °C, 30 s at 60 °C and 30 s at 72 °C. The PCR products were analyzed on 1% agarose gels containing 0.5 μg/ml ethidium bromide. The sequence of oligos is in Supplementary Table 4.

For qPCR experiments, RNA with RIN number >8 was used. GAPDH and 18S rRNA was used for normalization. A single reaction mix consisted of 0.5 μl cDNA, 0.5 μl primer mix, 2.5 μl SYBR green (applied biosystems, 4385612), and 1.5 μl water. Manufacturer of plates/tubes and catalog number: Thermo scientific—4309849. Thermocycling parameters: Hold Stage—95 °C for 3 min, PCR Stage—95 °C for 15 s; 60 °C for 30 s (45 cycles); Melt Curve Stage—95 °C for 15 s; 60 °C for 1 min; 95 °C for 15 s. Manufacturer of qPCR instrument: Applied Biosystems by Thermo Fisher Scientific. qPCR analysis program (source, version): QuantStudio

Design & Analysis Software v1.5.1. The specificity of the target was confirmed by a single sharp melt curve and resolving products on an agarose gel. RNA was isolated using Qiagen Kit and subjected to DNaseI treatment for 30 min at 37 °C. RNA was quantified using Qubit. Primers were designed using Primer Blast[75].

**Antibodies**. hnRNP L (CST 37562, dilution 1:3000), hnRNP LL (CST 4783S, dilution 1:3000), FLAG (Sigma-Aldrich A8592, dilution 1:10,000), Pol II Ser2P (Abcam ab5095, dilution 1:5000), Halo (Promega G9211, dilution 1:10,000), SETD2 (Aviva OAEB00589, dilution 1:3000), HA (Sigma 04-902, dilution 1:10,000), β-actin (Abcam ab8224, dilution 1:2500).

**High throughput sequencing**. Sequencing libraries were prepared using High Throughput Library Prep Kit (KAPA Biosystems) following the manufacturer's instructions. The library was sequenced on an Illumina HiSeq platform with paired reads of 75 bp for RNA-seq.

**RNA-seq analysis**. Raw reads were demultiplexed into FASTQ format allowing up to one mismatch using Illumina bcl2fastq2 v2.18. Reads were aligned to the human genome (hg38 and Ensembl 102 gene models) using STAR (version STAR_2.7.3a)[76]. TPM expression values were generated using RSEM (version v1.3.0). edgeR (version 3.24.3 with R 3.5.2) was applied to perform differential expression analysis, using only protein-coding and lncRNA genes[77]. To perform differential splicing analysis, we used rMATs (version 4.0.2) with default parameters starting from FASTQ files[78]. FDR cutoff of 0.05 was used to determine statistical significance.

**Reporting summary**. Further information on research design is available in the Nature Research Reporting Summary linked to this article.

## Data availability
All relevant data are available. The RNA-data sets are available in the Gene Expression Omnibus (GEO) database under the accession number GSE174426. The mass spectrometry proteomics data is available at the ProteomeXchange Consortium via the PRIDE partner repository with the dataset identifiers PXD019376, PXD022946, and PXD025942. The hnRNP L-SETD2 and the hnRNP LL-SETD2 complexes have been deposited to PDB with entry ID: 7EVR and 7EVS, respectively. The structure of PTB1-PRI3 with PDB ID: 3ZZY was used as the search model for determining hnRNP L-SETD2 complex. Source data are provided with this paper. Original data underlying this manuscript can be accessed from the Stowers Original Data Repository at http://www.stowers.org/research/publications/libpb-1662. Source data are provided with this paper.

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

## Acknowledgements

This work was supported by funding from the National Institute of General Medical Sciences (grant No. R35GM118068) and the Stowers Institute for Medical Research to Jerry L. Workman. This work was also financially supported by the grants from the Ministry of Science and Technology of China (2016YFA0500700 and 2019YFA0508403); Strategic Priority Research Program of the Chinese Academy of Sciences (XDB39000000); Chinese National Natural Science Foundation (32090040, 31870760, and U1932122). The authors would like to thank the members of the Workman lab for their critical suggestions to improve the paper. We thank the staff of BL17U1, BL18U1, and BL19U1 beamlines at the Shanghai Synchrotron Radiation Facility for assistance during X-ray diffraction data collection.

## Author contributions

S.B. conceptualized the work, designed, and performed the experiments. S.B. wrote the paper. S.W. performed ITC and crystallization experiments. S.W., S.S., Y.S. and F.L. analyzed the ITC and crystallography data and wrote the structural aspect of the work. D.R. generated the mutants and performed qPCR. Y.Z. and L.F. conducted mass spectrometry. Y.Z., L.F. and M.P.W. analyzed mass spectrometry data. N.Z. and H.L. analyzed the high-throughput sequencing data. F.L., Y.S. and J.L.W conceived the idea of the work, provided supervision, acquired funding, and revised the paper.

## Competing interests

The authors declare no competing interests.
