## [Peer Review File · Nature Communications]

Structural basis of the interaction between SETD2 methyltransferase and hnRNP L paralogs for governing co-transcriptional splicingReviewers' Comments:

Reviewer #1:

Remarks to the Author:

The work presented by Bhattacharya et al. is a detailed well written study that looks at the coupling of alternative splicing to transcription through the discovery that RRM2 of hnRNP L and LL bind to the methyltransferase SETD2 through the SHI domain. They present two new crystal complexes of the two hnRNP proteins with a SETD2 based peptide and used other experimental approaches MudPIT, ITC, and RNA-seq to explore the protein-protein interactions and the regulation to show that the function of hnRNP L and hnRNP LL are at least partially redundant.

I think this study is complete, and no additional experiments are needed only fixing of the errors I noted and consideration of my suggestions for improving readability of the paper. The authors presented a manuscript that is valuable to the field and should be published in Nature Communications.

Specific comments for the merged PDF file and supplementary material:

1. Page 6, line 110, Cite use of HMMER web search server
2. Page 6, line 111, Figure 1a has yellow text in sequence alignment that is hard to see, try another color.
3. Page 6, line 116, Please use the preferred name equilibrium dissociation constant K_{D} and stick with K_{D} as it is used in the ITC figures.
4. Supplementary Table 1 would be informative to reader is it included ΔG and preferably a K_{rel} to the appropriate WT for your many mutant ITC results. Also please state explicitly the number of repeats for your ITC experiments. If there are multiple runs please state the number, then show and state standard deviation or error for these average values where appropriate. Title should be more descriptive.
5. Page 7, line 131, If the black arrow is supposed to α -Pol II it could be marked as such or could be better aligned with Figure 1f. Please stick with the use of either α -Pol II or α -Pol II Se2 P so as not to be confusing, only one antibody is listed in methods.
6. Page 7, line 140, SETD2 2167-2192 is shown but Supplementary Table 2 has 2168-2192 this is confusing is residue 2167 is missing/disordered or is it a typo? If it is disordered still use the full length 2167-2192 peptide in the title. Same for hnRNP LL as it is SETD2 2181-2192 in the table but SETD2 2180-2192 everywhere else. Also Rampage, which is program/server, which is not cited, should be replaced with Ramachandran.
7. Page 8, line 145, please include what was used in the superimposition apo hnRNP L and the PDB ID, this also should be added to the figure legend for Supplementary Figure 1c.
8. Page 8, line 157, would be better served writing out 2190PTP2192 to something like 2190-Pro-Thr-Pro-2192. Labeling Tyr257 in Figure 2e would be helpful to readers.
9. Page 10, line 195, replace the confusing SETD2 Δ 2164-2213 with something closer to what was used in other sections like SETD2 Δ SHI²¹⁶⁴⁻²²¹³
10. Page 10, line 212, Please add the percentage for the sequence homology between hnRNP L and LL.
11. Page 12, line 219, First use of "FL" in hnRNP LL FL (1-542) so full length should be written before

the abbreviation so as not to confuse the reader.

12. Page 12, line 248, there is no need for figure 5b as it has everything that is 5a. Figure 5a could be improved by coloring the line to match the symbol.

13. Page 13, line 279, PTB ITC experiment was run but results were not included Supplementary Table 1.

14. Page 30, line 773, please add citations for the servers/programs ESPript3 and CLUSTALW.

15. Page 31, line 795, extra (a) is not needed.

Reviewer #2:

Remarks to the Author:

In this manuscript, Bhattacharya, Wang et al. present evidence for a mechanism by which hnRNP L/LL can interact with their partner SETD2. The findings are important in that they unveil a mechanism that might very well be common to several other RNA-binding proteins and provide additional information on how target RNA binding specificity can be achieved for RBPs that have short/degenerated binding motifs. The authors do a remarkable job of making clear what protein regions and mutants are used in each experiment and in what context each experiment is performed. They also report rigorous and strong experimental evidence to support their claims. Overall, this is a strong manuscript that flows well and provides important mechanistic insight into the interplay between RBPs and their RNA targets and protein partners.

My main concern lies with the quantification of exon inclusion in TJP1 and BPTF in Figure 6. The use of RT-qPCR with alternative exon-specific primers, as shown in Figure 6, is not and will never be an acceptable way to measure isoform ratios. For this purpose, semi-quantitative RT-PCR with primers annealing in flanking exons must be used to allow for the visualization of both the included and skipped isoforms on the same lane and from a single reaction. The quantification of exon inclusion/skipping in TJP1 and BPTF need to be repeated using this method to determine the extent of the rescue by the hnRNP L/LL paralogs.

Additional comments:

I do not see the purpose of Figure 5B, which, from my understanding, is a copy of 5A with a few curves removed. If this really is the case, this panel must be removed.

In the section relating to L and LL redundancy, the authors state that "overlap between SETD2 and hnRNP LL regulated gene expression and AS events was very small" but do not show whether the depletion in the overlap is statistically significant.

The use of a single reference gene for normalization of RT-qPCR data (GAPDH in this case) is poor practice. Please refer to MIQE guidelines.

Legend to Figure 6A mentions that "Western blot of whole-cell lysates was performed with the depicted antibodies" but no Western is shown in Figure 6.

Some comments push too far in their claims. E.g. in the intro: "Their [hnRNPs] function during AS is to bind to the ESS and exclude the SR proteins." Although hnRNPs can compete with SR proteins, they don't always do and claiming that it is their role to exclude SR proteins is an overreach.

Typos/editing make a sentence hard to understand in the intro: "Med23 brings hnRNP L to the promoter of target genes from where it might be is handed over to SETD2 to co-regulate a common

subset of AS events”.

Spelling out “leucine-leucine pair” or “Leu-Leu pair” rather than using “LL pair” everywhere in the text would avoid confusion between references to the leucine-leucine pair of SETD2 and hnRNP LL (so many LLs!).

I would have liked to see a short word on the roles of SETD2 and MED23 in the intro. I also think that properly introducing LL would help emphasizing the importance of results presented in Figures 3-5: how much was previously known about whether it compete with L, are they redundant, are they expressed in a mutually exclusive manner, do they bind different targets, does LL do anything important on its own?

Reviewer #3:

Remarks to the Author:

The authors are presenting here an interesting structural study between the alternative-splicing factors hnRNPL and LL with the SETD2 methyltransferase. Interactions between hnRNP L and SETD2 was found previously by this group but the molecular basis of the interaction was not determined. here they found that a small disordered region of SETD2 interact with the RRM2 of hnRNP L and LL and could determine the 3D structure of this interaction. Similarly to other RRMs like PTB RRM2 or UHM-like RRM, SETD2 interact with a cavity between the two helix of the RRM away from the RNA binding surface. The structure is reminiscent of the PTB RRM2 bound to Raver-1. The structure is interesting as it connect a transcription factor and a splicing factor supporting the coupling between transcription and splicing. The structural data, biochemical data are both convincing. Major point:

The functional data are less well connected with the structural work. One would have expected to see if for example protein mutants of hnRNP L or LL that still can bind RNA but not SETD2 would have a different role in splicing or maybe different RNA binding targets. If such functional data were performed, this would strenghten the functional relevance of the structure and the paper in general. Minor point:

It is not clear how the 3D model of the complex with RNA has been generated in figure S1.

NCOMMS-21-24008-T

Structural basis of the interaction between SETD2 methyltransferase and hnRNP L paralogs for governing co-transcriptional splicing

Saikat Bhattacharya, Suman Wang, Divya Reddy, Siyuan Shen, Ying Zhang, Ning Zhang, Hua Li, Michael Washburn, Laurence Florens, Yunyu Shi, Fudong Li and Jerry L. Workman

The authors would like to thank the reviewers for providing valuable feedback and suggestions to improve our manuscript, and the editor, for allowing us an opportunity to submit a revised version. The figures, their legends, and the manuscript text have been modified to incorporate the suggestions. The modified text has been highlighted in yellow in the revised manuscript.

REVIEWER COMMENTS

Reviewer #1 (Structural biology of RNA and RNA-Protein complexes):

The work presented by Bhattacharya et al. is a detailed well written study that looks at the coupling of alternative splicing to transcription through the discovery that RRM2 of hnRNP L and LL bind to the methyltransferase SETD2 through the SHI domain. They present two new crystal complexes of the two hnRNP proteins with a SETD2 based peptide and used other experimental approaches MudPIT, ITC, and RNA-seq to explore the protein-protein interactions and the regulation to show that the function of hnRNP L and hnRNP LL are at least partially redundant.

I think this study is complete, and no additional experiments are needed only fixing of the errors I noted and consideration of my suggestions for improving readability of the paper. The authors presented a manuscript that is valuable to the field and should be published in Nature Communications.

- *The authors would like to thank the reviewer for the positive remarks and constructive suggestions. These have helped us in improving the overall quality of our manuscript.*

1. Page 6, line 110, Cite use of HMMER web search server

- *Thank you for pointing out the missing citation. Citations has been added in the revised manuscript.*

2. Page 6, line 111, Figure 1a has yellow text in sequence alignment that is hard to see, try another color.

- *We have changed the text to black.*

3. Page 6, line 116, Please use the preferred name equilibrium dissociation constant K_D and stick with K_D as it is used in the ITC figures.

- *We have incorporated the change suggested by the reviewer.*

4. Supplementary Table 1 would be informative to reader is it included ΔG and preferably a K relative (K_{rel}) to the appropriate WT for your many mutant ITC results. Also please state explicitly the number of repeats for your ITC experiments. If there are multiple runs please state the number, then show and state standard deviation or error for these average values where appropriate. Title should be more descriptive.

- *We would like to thank the reviewer for the suggestion. The revised Supplementary Data 1 now includes ΔG and K_{rel} , as well as the average K_D Value and standard deviation of each three repeats. Also, we have included all the ITC curves in a Supplementary Data 2 of the revised manuscript. The title of the ITC plots in main and supplementary figures have been made more descriptive.*

5. Page 7, line 131, If the black arrow is supposed to α -Pol II it could be marked as such or could be better aligned with Figure 1f. Please stick with the use of either α -Pol II or α -Pol II Ser2 P so as not to be confusing, only one antibody is listed in methods.

- *The arrow in Figure 1e is indicating hnRNP L. It has now been marked as such in the revised manuscript. The antibody used was against α -RNA Pol II Ser2 P. We thank the reviewer for pointing out this error. The labeling has been corrected in all the figures.*

6. Page 7, line 140, SETD2 2167-2192 is shown but Supplementary Table 2 has 2168-2192 this is confusing is residue 2167 is missing/disordered or is it a typo? If it is disordered still use the full length 2167-2192 peptide in the title. Same for hnRNP LL as it is SETD2 2181-2192 in the table but SETD2 2180-2192 everywhere else. Also Rampage, which is program/server, which is not cited, should be replaced with Ramachandran.

- *Yes, the density of residue 2167 could not be seen in the final map. We have modified "2168-2192" in Supplementary table to "2167-2192" in the title. We also modified "2181-2192" to "2180-2192".*
- *As per reviewer's suggestion, we have changed "Rampage" to "Ramachandran" in the revised manuscript. RAMPAGE is an offshoot of RAPPER which generates a Ramachandran plot using data derived by the Richardson and coworkers. It is recommended that it be used for this purpose in preference to PROCHECK, which is based on much older data. The RAMPAGE program is supported in Phenix. Hence, we have cited Richardson's paper and phenix.*

7. Page 8, line 145, please include what was used in the superimposition apo hnRNP L and the PDB ID, this also should be added to the figure legend for Supplementary Figure 1c.

- *We would like to thank the reviewer for pointing out the missing information from our manuscript. Mouse apo hnRNP L RRM2 (PDB ID: 2MQM), that has 100% sequence identity with human hnRNP L RRM2, was used in the superimposition. This information has now been included in the revised manuscript text and figure legend.*

8. Page 8, line 157, would be better served writing out 2190PTP2192 to something like 2190-Pro-Thr-Pro-2192. Labeling Tyr257 in Figure 2e would be helpful to readers.

- *We have changed 2190PTP2192 to 2190-Pro-Thr-Pro-2192 in the revised manuscript text. Tyr257 has now been labelled in the revised figure 2e.*

9. Page 10, line 195, replace the confusing SETD2CΔ2164-2213 with something closer to what was used in other sections like SETD2CASHI²¹⁶⁴⁻²²¹³

- *We have made the suggested change in the revised manuscript.*

10. Page 10, line 212, Please add the percentage for the sequence homology between hnRNP L and LL.

- *We have added the information that hnRNP L and LL are 71% homologous in the revised manuscript.*

11. Page 12, line 219, First use of “FL” in hnRNP LL FL (1-542) so full length should be written before the abbreviation so as not to confuse the reader.

- *We have made the suggested change in the revised manuscript.*

12. Page 12, line 248, there is no need for figure 5b as it has everything that is 5a. Figure 5a could be improved by coloring the line to match the symbol.

- *As per reviewer’s suggestion, we have removed the figure 5b of the original submission. Also, we have matched the color of the line and symbol of Figure 5a.*

13. Page 13, line 279, PTB ITC experiment was run but results were not included Supplementary Table 1.

- *We would like to thank the reviewer for pointing out the missing data from the manuscript. The data of PTB ITC experiments have now been included in the Supplementary Data 1 and 2 of the revised manuscript.*

14. Page 30, line 773, please add citations for the servers/programs ESPript3 and CLUSTALW.

- *Thank you for pointing out the missing citations. Citations for these tools have been added in the revised manuscript.*

15. Page 31, line 795, extra (a) is not needed.

- *This correction have been incorporated in the revised manuscript.*

Reviewer #2 (Alternative splicing):

In this manuscript, Bhattacharya, Wang et al. present evidence for a mechanism by which hnRNP L/LL can interact with their partner SETD2. The findings are important in that they unveil a mechanism that might very well be common to several other RNA-binding proteins and provide additional information on how target RNA binding specificity can be achieved for RBPs that have short/degenerated binding motifs. The authors do a remarkable job of making clear what protein regions and mutants are used in each experiment and in what context each experiment is performed. They also report rigorous and strong experimental evidence to support their claims. Overall, this is a strong manuscript that flows well and provides important mechanistic insight into the interplay between RBPs and their RNA targets and protein partners.

- *The authors would like to thank the reviewer for the positive remarks and constructive suggestions. These have helped us in strengthening our work and improving the overall quality of our manuscript.*

1. My main concern lies with the quantification of exon inclusion in TJP1 and BPTF in Figure 6. The use of RT-qPCR with alternative exon-specific primers, as shown in Figure 6, is not and will never be an acceptable way to measure isoform ratios. For this purpose, semi-quantitative RT-PCR with primers annealing in flanking exons must be used to allow for the visualization of both the included and skipped isoforms on the same lane and from a single reaction. The quantification of exon inclusion/skipping in TJP1 and BPTF need to be repeated using this method to determine the extent of the rescue by the hnRNP L/LL paralogs.

- *As suggested by the reviewer, RT-PCR was performed to allow for the visualization of both the included and skipped isoforms of tjp1 and bptf on the same lane (Figure 1). This data has now been included in figure 6 of the revised manuscript in addition to the quantitative PCR data that doesn't suffer from the drawback of trying to quantify saturated bands from gel images while trying to quantify isoform ratios.*

Figure 1. RT-PCR products resolved on agarose gel showing the isoforms of *tjp1* and *bptf*.

Additional comments:

2. I do not see the purpose of Figure 5B, which, from my understanding, is a copy of 5A with a few curves removed. If this really is the case, this panel must be removed.

- *As per reviewer's suggestion, we have removed the figure 5b of the original submission.*

3. In the section relating to L and LL redundancy, the authors state that “overlap between SETD2 and hnRNP LL regulated gene expression and AS events was very small” but do not show whether the depletion in the overlap is statistically significant.

- *We would like to thank the reviewer for suggesting changes that will make it easier for the readers to understand our results. All the gene expression (FDR<0.05, fold change>1.5) and*

AS changes (FDR<0.05) discussed in the manuscript are statistically significant. Also, we have now added a sentence in the manuscript text mentioning the number of AS events that overlap between SETD2 and hnRNP LL.

4. The use of a single reference gene for normalization of RT-qPCR data (GAPDH in this case) is poor practice. Please refer to MIQE guidelines.

- *As per reviewer's suggestion, in addition to GAPDH, 18S rRNA was also used as a normalization control. This data has been included as Supplementary Figure 6 of the revised manuscript. Also, information regarding controls, method, instrumentation etc. has been included in the revised manuscript as per MIQE guidelines.*

5. Legend to Figure 6A mentions that "Western blot of whole-cell lysates was performed with the depicted antibodies" but no Western is shown in Figure 6.

- *We would like to thank the reviewer for pointing out this error in our original submission. This has been corrected in the revised manuscript.*

6. Some comments push too far in their claims. E.g. in the intro: "Their [hnRNPs] function during AS is to bind to the ESS and exclude the SR proteins." Although hnRNPs can compete with SR proteins, they don't always do and claiming that it is their role to exclude SR proteins is an overreach.

- *Keeping in mind the reviewer's suggestion, we have modified the text in the revised manuscript.*

7. Typos/editing make a sentence hard to understand in the intro: "Med23 brings hnRNP L to the promoter of target genes from where it might be is handed over to SETD2 to co-regulate a common subset of AS events".

- *We apologize for the typing error. We have corrected that and also, performed a thorough proofreading of the entire manuscript.*

8. Spelling out "leucine-leucine pair" or "Leu-Leu pair" rather than using "LL pair" everywhere in the text would avoid confusion between references to the leucine-leucine pair of SETD2 and hnRNP LL (so many LLs!).

- *That is a very useful suggestion. We agree that were too many LLs in the manuscript and that can be confusing at times. We have replaced LL pair with Leu-Leu in text as well as the figures of the revised manuscript.*

9. I would have liked to see a short word on the roles of SETD2 and MED23 in the intro. I also think that properly introducing LL would help emphasizing the importance of results presented in Figures 3-5: how much was previously known about whether it compete with L, are they redundant, are they expressed in a mutually exclusive manner, do they bind different targets, does LL do anything important on its own?

- *We would like to thank the reviewer for constructive suggestion to improve our manuscript. The introduction section of the revised manuscript has been modified to better introduce the readers to SETD2, Med23 and hnRNP LL.*

Reviewer #3 (Structural biology, biochemistry, RNA-protein mode of interaction):

The authors are presenting here an interesting structural study between the alternative-splicing factors hnRNPL and LL with the SETD2 methyltransferase. Interactions between hnRNP L and SETD2 was found previously by this group but the molecular basis of the interaction was not determined. here they found that a small disordered region of SETD2 interact with the RRM2 of hnRNP L and LL and could determine the 3D structure of this interaction. Similarly to other RRMs like PTB RRM2 or UHM-like RRM, SETD2 interact with a cavity between the two helix of the RRM away from the RNA binding surface. The structure is reminiscent of the PTB RRM2 bound to Raver-1.

The structure is interesting as it connect a transcription factor and a splicing factor supporting the coupling between transcription and splicing. The structural data, biochemical data are both convincing.

- *The authors would like to thank the reviewer for the positive remarks.*

Major point:

The functional data are less well connected with the structural work. One would have expected to see if for example protein mutants of hnRNP L or LL that still can bind RNA but not SETD2 would have a different role in splicing or maybe different RNA binding targets. If such functional data were performed, this would strengthen the functional relevance of the structure and the paper in general.

- *We would like to thank the reviewer for the constructive criticism of our work. The authors agree with the reviewer's comments. However, technical difficulties have impeded our efforts to perform experiments that would allow us to address those comments.*
 - *To understand whether hnRNP L/LL mutants, that can bind RNA but not SETD2, have different RNA binding targets, we tried performing RIP-Seq of hnRNP L using a commercially available kit (Active Motif RNA ChIP-IT® Kit #53024). The RRMs of hnRNPs including their overall sequence can be very similar. Hence, it is critical to use antibody specific for a given hnRNP, especially in experiments like ChIP or RIP to draw meaningful conclusions. We used the CST #37562 hnRNP L antibody in our experiments. The antigen sequences of CST antibodies are proprietary; however, we were informed by CST that the #37562 hnRNP L antibody was produced by immunizing animals with a synthetic peptide corresponding to residues surrounding His313 of human hnRNP L protein. Based on BLAST alignment results, there is a high likelihood that this antibody might be specific for hnRNP L and will not*

Figure 2. Bar graph showing an enrichment of Suz12 at the SFPQ locus in RIP experiment by qRT-PCR.

bind hnRNP LL. The Product Usage page on the CST website does not list CHIP and RIP as applications for this antibody. Unfortunately, our multiple attempts to perform hnRNP L CHIP- and RIP-Seq did not yield much success. Upon sequencing of the ChIPed DNA, specific enrichment of hnRNP L was not observed although more than 25 million reads aligned to the genome. Library preparation did not succeed in our RIP experiments. Notably, RIP of the positive control Suz12 (supplied by Active Motif RNA ChIP-IT® Control Kit – Human #53025) worked as expected and a clear enrichment was observed at the SFPQ locus in qRT-PCR experiments [Figure 2]. It is likely that the antibodies against hnRNP L used by us were not RIP-grade, resulting in our failure to pull down hnRNP L effectively. The possibility of a promiscuous cross-linking of hnRNP L to chromatin makes it an unsuitable candidate for performing epitope-tagged CHIP or RIP.

○ Our experiments have revealed that hnRNP L and LL are at least partially redundant. Hence, to definitively test whether SETD2-non-binding mutant of hnRNP L/LL have a different effect on splicing, we need a cell line in which both the paralogs are deleted or depleted. Mutant hnRNP L/LL proteins can then be introduced in this cell line to address that question. So far, we haven't succeeded in generating such a cell line using CRISPR. We are currently exploring whether inducible knockdown will allow us to generate such a system. Our current manuscript, the focus of which is the molecular basis of SETD2-hnRNP L interaction, provides important evidence of a conserved design by which RRM engage their protein interactors. This information, along with a suitable model system will allow us to strengthen the importance of our structural findings with functional outcomes going forward.

Minor point:

It is not clear how the 3D model of the complex with RNA has been generated in figure S1.

- We would like to thank the reviewer for pointing out the missing information from our manuscript. The RRM-RNA model is the structure of hnRNP L RRM1 in complex with a 5'-CACAC-3' RNA sequence (PDB ID: 2MQO). Most of the RRMs bind to RNA in the same beta-sheet face but not dorsal helix face. The superimposition of the two structure clearly show that the peptide binding and RNA binding interfaces are separated. This information has now been included in Supplementary Figure 1c legend.*

Reviewers' Comments:

Reviewer #1:

Remarks to the Author:

I believe Bhattacharya et al. have made the changes requested by the reviewers or given reasonable explanations of why it was not feasible to do so, thus bolstering the experimental evidence and improving the readability of the paper. My one comment is they can reduce the instances of the word Exemplary in the figure legends to describe their ITC data, one use in the legend of Figure 1. is enough. The paper is ready for publication in Nature Communications.

Reviewer #2:

Remarks to the Author:

The authors did a great job with the rebuttal. My comments have been satisfactorily addressed.

NCOMMS-21-24008A

Structural basis of the interaction between SETD2 methyltransferase and hnRNP L paralogs for governing co-transcriptional splicing

Saikat Bhattacharya, Suman Wang, Divya Reddy, Siyuan Shen, Ying Zhang, Ning Zhang, Hua Li, Michael Washburn, Laurence Florens, Yunyu Shi, Jerry L. Workman and Fudong Li

REVIEWER COMMENTS

Reviewer #1 (Remarks to the Author):

I believe Bhattacharya et al. have made the changes requested by the reviewers or given reasonable explanations of why it was not feasible to do so, thus bolstering the experimental evidence and improving the readability of the paper. My one comment is they can reduce the instances of the word Exemplary in the figure legends to describe their ITC data, one use in the legend of Figure 1. is enough. The paper is ready for publication in Nature Communications.

- *The authors once again would like to thank the reviewer for the constructive feedback on our manuscript for its improvement. We have made the changes in the figure legends that was asked by the reviewer.*

Reviewer #2 (Remarks to the Author):

The authors did a great job with the rebuttal. My comments have been satisfactorily addressed.

- *The authors would like to thank the reviewer for the positive remarks and constructive suggestions. These have helped us in strengthening our work and improving the overall quality of our manuscript.*